# Soup to go: mitigating forgetting during continual learning with model averaging

## Abstract

In continual learning, where task data arrives in a sequence, fine-tuning on later tasks will often lead to performance degradation on earlier tasks. This is especially pronounced when these tasks come from diverse domains. In this setting, how can we mitigate catastrophic forgetting of earlier tasks and retain what the model has learned with minimal computational expenses? We propose Sequential Fine-tuning with Averaging (SFA), a method that merges currently training models with earlier checkpoints *during the course of training*. Our method outperforms SOTA merging, and penalty methods, and achieves comparable performance to rehearsal with just a data buffer. In turn, our method offers insight into the benefits of merging partially trained models during training across both image and language domains.

## 1 Introduction

Fine-tuning deep learning models on new tasks often leads to catastrophic forgetting: the rapid degradation of performance on previously learned tasks (Scialom et al., 2022; Lesort et al., 2019; Delange et al., 2021; Belouadah et al., 2021; Luo et al., 2023). This poses a major challenge for continual learning (CL) scenarios, where data comes in a stream of sequences of tasks that may not reappear. As such, we are in need of fine-tuning procedures that would allow models to continually adapt to new knowledge without sacrificing past abilities.

Previous work has analyzed catastrophic forgetting of different types of information, as well as the impact of scale. Scialom et al. (2022) explain that LLMs can perform worse on past fine-tuning tasks as they learn new ones. Furthermore, Luo et al. (2023) show a model can also forget general knowledge, not specific to a single past task.

Existing state-of-the-art approaches to mitigate forgetting primarily focus on modifying the training data used in fine-tuning. These methods either maintain a data buffer of past tasks (Robins, 1995; Lopez-Paz & Ranzato, 2022; de Masson d'Autume et al., 2019), or generate approximations of past task data for joint training with current tasks (Shin et al., 2017; Mocanu et al., 2016). However, both strategies introduce additional costs. Data buffers increase memory overhead and require careful management, while generating data approximations necessitates extra training and computational resources. Likewise, more classical methods of CL that incorporate a penalty directly into training to constrain weights ((Kirkpatrick et al., 2017), L2 penalty) are memory-intensive as they require storing multiple copies of model parameters to be used at each gradient step.

Recently, buffer-free and computationally efficient model merging techniques (Wortsman et al., 2022b; Ilharco et al., 2023; Marouf et al., 2024) have been proposed to address forgetting in CL (Roth et al., 2024; Kozal et al., 2024; Marczak et al., 2024). However, in scenarios involving numerous tasks or domains with significant variation, these methods often struggle to achieve a competitive balance between retaining knowledge of previous tasks and learning new ones. Furthermore, while model merging during training to achieve this balance has been explored in prior work, existing approaches typically rely on additional buffers, auxiliary models, or explicit regularizers, and often perform averaging at every iteration—substantially increasing computational and memory costs (Döbler et al., 2023; Soutif-Cormerais et al., 2023; Sarfraz et al., 2023; Bhat et al., 2024; Sarfraz et al., 2022; Gowda et al., 2023; Arani et al., 2022; Tarvainen & Valpola, 2018). To address

this gap, we propose **Sequential Fine-tuning with Averaging (SFA)**, a method that merges the model currently fine-tuning on a new task with a checkpoint from a prior task *during* training, without requiring replay buffers or auxiliary regularizers. Unlike approaches that merge continuously, SFA introduces a tunable averaging frequency parameter ($p$) that governs how often merging occurs, enabling a principled balance between retaining past knowledge and adapting to new tasks. Our solution also does not require training an additional past data generator, because it uses previous model checkpoints as proxies for such data. Finally, our experiments demonstrate that our method, by incorporating averaging during training, outperforms other merging methods that only merge at the conclusion of training, achieving superior performance across all tasks. We systematically investigate forgetting across two extensive settings: (1) fine-tuning pretrained large language models (LLMs) on highly distinct domains, including Law, Math, and Code, and (2) fine-tuning image models on a large sequence of classification tasks. These settings were selected to evaluate SFA's robustness in two key scenarios: one involving drastic domain shifts between tasks, and another involving long sequences of tasks during extended continual learning. Our work can be summarized by the following contributions:

- We introduce Sequential Fine-tuning with Averaging (SFA), a method for mitigating forgetting by averaging model checkpoints from past tasks during fine-tuning on a new task. This enables the model to retain knowledge on past tasks while learning a new task (Section 3).
- We show consistent results that across a scale of models and for both language and image tasks, SFA outperforms other model merging techniques, as well as classical continual learning methods (Sections 4 and 5).
- We provide intuition for why model merging is effective by showing how SFA roughly approximates a classical continual learning algorithm: L2-regression. In turn, we bridge classical continual learning algorithms that incorporate a penalty with commonly used model merging techniques. (Section 5).

## 2 Related Work

**Forgetting and Continual Learning**  A large and growing body of literature investigates different aspects of catastrophic forgetting in continual and sequential learning. When the training data consists of disjoint tasks, training classifiers can cause catastrophic forgetting (Rebuffi et al., 2017). Furthermore, if forgetting occurs, it can be tracked during training and is dependent on when examples are seen by the model: models are less likely to remember earlier training examples (Jagielski et al., 2022; Tirumala et al., 2022). Interestingly, forgetting can also occur for general knowledge rather than for specific tasks, and is more severe for larger models (Luo et al., 2023). Lesort et al. (2022) show that overlap between tasks and task repetition in continual learning settings can mitigate catastrophic forgetting of such examples resulting in solutions to forgetting that involve maintaining a data buffer with past data. This may also explain why Rehearsal sampling is one of the most popular and effective continual learning solutions (Prabhu et al., 2023; Garg et al., 2024). Furthermore, certain variations of data stream repetition may make solutions like ensembling especially effective (Hemati et al., 2025). These findings can also be extrapolated to LLMs where continual learning with data repetition can prevent catastrophic forgetting (Scialom et al., 2022). Mitigating forgetting in continual learning can also occur by introducing a penalty in the loss objective. L2 penalty in continual learning constrains the weights of a model as it is learning a new task by introducing a penalty based on the difference between the current and initial model's weights. Similarly, another popular method Elastic Weight Consolidation (EWC) (Kirkpatrick et al., 2017; van de Ven, 2025) also introduces a penalty to constrain the weights of a model and mitigate increased loss on learned tasks while incorporating the importance of specific weights on learned tasks. Finally, weight averaging in continual learning has been explored in the literature. One such line of work employs continuous averaging strategies based on exponential moving averages (EMA). Robust Mean Teacher (RMT) adopts a mean-teacher framework to adapt models continually at test time (Döbler et al., 2023). Soutif-Cormerais et al. (2023) likewise maintain an EMA of parameters to improve stability in online continual learning (Soutif-Cormerais et al., 2023). However, unlike SFA which incorporates infrequent averaging to mitigate costs, these methods rely on averaging at each iteration. Given scaling models, such implementation can be costly in terms of memory and compute. Another major line of work relies on episodic buffers to replay past data and mitigate forgetting, in combination with averaging mechanisms. ESMER couples its stable EMA model with error-sensitive reservoir sampling Sarfraz et al.

(2023); IMEX-Reg supplements rehearsal with regularizers Bhat et al. (2024); SCoMMER combines a replay buffer with activation sparsity, semantic dropout, and long-term memory Sarfraz et al. (2022); and DUCA integrates buffer-based replay with its multi-memory system Gowda et al. (2023). However, as explained above, maintaining a buffer can be costly and requires additional storage. In contrast, SFA only relies on the periodic merging of two model checkpoints with no additional buffer. Finally, CLS-ER (Arani et al., 2022), a biologically inspired framework, builds two *semantic memories*—a fast plastic model and a slow stable model—updated at different frequencies, and couples them with an episodic buffer populated by reservoir sampling; a consistency loss nudges the working model toward whichever semantic memory better supports the current label, yielding strong results in streaming settings. On the other hand, SFA dispenses with any data buffer, auxiliary teacher models and extra forward passes: during fine-tuning on a new task, it periodically averages the current parameters with the prior optimal checkpoint (only storing two model parameters, as opposed to three). This can lead to significant storage drops especially for larger models or training runs (Appendix A.11). Furthermore, in comparison to many of these methods which only evaluate in the vision domain, SFA successfully extends continual learning with parameter averaging to a language task setting. Thus, when data retention is constrained (privacy, licensing) or minimal overhead is paramount, SFA's in-training averaging offers a path to a past and new task learning balance.

**Model Merging** There exist many techniques and applications for merging multiple models to create a single model with improved generalization on a given set of tasks. Model souping (Wortsman et al., 2022a) involves averaging the parameters of existing models to create a new model. This technique can be applied after training multiple variations of a model on data during a hyperparameter sweep to combine the models and achieve higher performance than any individual model. Task Arithmetic (Ilharco et al., 2023) involves finding and adding task vectors to create a multi-task model. WiSE-FT (Wortsman et al., 2022b) merges the weights of a pretrained and a fine-tuned model. Our method builds upon these works, but with key differences as described in Section 3.

Additional influential model merging techniques include: Ramé et al. (2023) use a model souping approach to obtain a network with improved out-of-distribution performance by averaging the weights of models fine-tuned on different tasks. TIES (Yadav et al., 2023) only merges influential parameters whose signs are in the direction of greatest movement across the models. Fisher merging (Matena & Raffel, 2022; Dhawan et al., 2023; Jhunjhunwala et al., 2023) requires keeping data from all previous tasks and computing gradients. In continual learning specifically, approaches that range from using momentum-based weight interpolation (Stojanovski et al., 2022), to merging an existing and a fine-tuned model (Ilharco et al., 2022; Marouf et al., 2024; Liu & Soatto, 2023) exist. Furthermore, methods that focus on merging with respect to task boundaries, rather than within task-training appear (Kozal et al., 2024; Marczak et al., 2024).Finally for merging different textual domains, Branch-Train-Merge (BTM) (Li et al., 2022) maintains a set of distinct domain models that can be merged and then trained to create new experts.

## 3 Methodology: Sequential Fine-tuning with Averaging (SFA)

Our method, Sequential Fine-tuning with Averaging (SFA), leverages existing techniques in model merging (Ilharco et al., 2023; Wortsman et al., 2022a;b), and L2-regression (Section 5) to mitigate forgetting in the continual learning setting. Furthermore, it also draws upon promising results from CLS-ER (Arani et al., 2022) which features in-training parameter averaging. In this method, we consider a pretrained model that is fine-tuning on a sequence of tasks or domains. While the model is being fine-tuned on the current task, we periodically average the parameters of the current model with an earlier checkpoint that resulted from fine-tuning on previous tasks. We then continue fine-tuning this new averaged model on the current task.

More precisely, let $\theta_0$ denote the parameters of the network optimized for previous tasks. Let $\theta_{t+1}^*$ be the parameters of the model after taking a gradient step on a new task at time $0 \leq t \leq T - 1$ using current model parameters $\theta_t$. Then, every $pT$ iterations, as well as at the end of fine-tuning, we reset the parameters to be a weighted combination of $\theta_0$ and $\theta_{t+1}^*$, where the weighing is determined by a hyperparameter $0 \leq \beta \leq 1$ (default: 0.5).

By averaging with an optimized model of the previous tasks $\theta_0$, our method prevents the current model parameters from moving significantly from the original model's and thus losing optimal performance on past

---

**Algorithm 1** Sequential Fine-tuning with Averaging (SFA)

---

**Input:** $\theta_0, p, \beta, T$
**for** $t$ in $0, ..., T - 1$
    $\theta_{t+1}^* = \theta_t - \alpha \nabla_{\theta_t} L_{\text{task}}$
    **if** $(t + 1) \mod pT = 0$ **then**
        $\theta_{t+1} = (\beta)\theta_0 + (1 - \beta)\theta_{t+1}^*$
    **else**
        $\theta_{t+1} = \theta_{t+1}^*$
**if** $T \mod pT \neq 0$ **then**
    $\theta_T^* = (\beta)\theta_0 + (1 - \beta)\theta_T$
**else**
    $\theta_T^* = \theta_T$

---

tasks (Appendix A.7). In this way, our technique combines the intuition of continual learning with Rehearsal (Robins, 1995), Task Arithmetic (Ilharco et al., 2023), and WiSE-FT (Wortsman et al., 2022b). However, unlike Rehearsal-based methods that store data in a buffer, we use a model fine-tuned on past tasks/domains. Furthermore, unlike Task Arithmetic, our method merges a past checkpoint of a given model with the current model, rather than the task vectors from individual models. Also, while our method focuses on merging during actual fine-tuning and across tasks/domains, WiSE-FT merges a pretrained and a fully fine-tuned model.

Finally, after finishing fine-tuning with SFA on a new task, we update $\theta_0$ to be the final merged model of all tasks seen so far. As such, SFA ($p = 1$) can be considered to be a generalized WiSE-FT across fine-tuning as it combines an initial and fine-tuned model. However, unlike WiSE-FT, SFA extends to merging during training itself.

## 4 Results

**Datasets and Models:** In order to test the efficacy of our method and compare it to other baselines, we fine-tune our models on a variety of datasets. Specifically, we use multiple tasks from 3 distinct language domains: Law, Math and Code, as well as, streams of image classification tasks from Food-101 (Bossard et al., 2014), and CIFAR-100 (Krizhevsky, 2009). Details about which datasets we use in our language domains, and how we construct the image tasks are in Appendix A.4. Finally, fine-tuning model descriptions, and details about hyperparameter selection can be found in Appendix A.1 and Appendix A.2 respectively.

**Evaluation Metrics for Data:** In our work, we reference the *forgetting* of various tasks. We define forgetting specific knowledge as a decrease in performance on a given evaluation task between the current model and the original model before fine-tuning. For example, if evaluation performance on Task A drops when a model fine-tunes on Task B, given that the model has already fine-tuned on task A, we consider the model to forget Task A. To evaluate performance on our fine-tuning data, we use the metrics and holdout sets described in Table 1. In summary, we focus on accuracy when evaluating performance on holdout sets, as it is a common metric in both continual learning and model merging (Garg et al., 2024; Wortsman et al., 2022b; Scialom et al., 2022). This helps bridge the gap between the two fields and clearly highlight performance differences.

### 4.1 Mitigating Forgetting from Cross Language Domains

We begin by evaluating our method on language tasks, where robust LLM systems and models that can effectively retain and leverage prior knowledge are the increasingly important desiderata for real-world settings. To this end, we focus on evaluating SFA's performance on pairs of successive instruction fine-tuning tasks with significant domain shifts—such as transitioning from Math to Code or Math to Law—using the datasets outlined in Appendix A.4. By restricting ourselves to pairs of dissimilar tasks, we can clearly quantify the trade off between learning the second task and forgetting the first one by visualizing the results on a plane that measures the accuracy of the first task on the y-axis and the accuracy of the second task on the x-axis.

First, we confirm that forgetting occurs when fine-tuning on successive language tasks (Appendix A.9). We present our results for sequentially learning Math and Law with Llama 2 (7B) in Figure 1, Math and Law with Qwen2.5 (1.5B) in Figure 2, and Math and Law, as well as Math and Code with Pythia (2.8B) in Figure 11.

We first fine-tune our model Llama 2 (7B) in Figure 1, Qwen2.5 (1.5B) in Figure 2, and Pythia (2.8B) in Figure 11) on MetaMathQA to obtain the inital model (dark blue circle). Note the base model performance on the first (second) task is represented by dark green for Llama 2 (7B), dark red for Qwen2.5 (1.5B), and blue for Pythia (2.8B) horizontal (vertical) dashed lines. This initial model improves upon the base model on our Math benchmark and is thus higher on the y-axis (performance on first task) while not being significantly different or being worse on the x-axis (performance on the second task which it has not been trained on yet). We then fine-tune the initial model on the second task to obtain the sequential fine-tuning model (red circle). In Figures 1 and 2 the second task is Law while in Figure 11 the second task is either Law or Code. The sequential fine-tuning model performs really well on the second task (higher on the x-axis) while forgetting almost everything it has learned about the first task (base model level on the y-axis). This movement down and to the right of the initial model (dark blue circle) to the sequential fine-tuning model (red circle) on the task 1 - task 2 performance plane in Figures 1, 2 and 11 is emblematic of catastrophic forgetting of an earlier task as the model learns a new task. For reference, the performance of just fine-tuning the base model on the second task is represented by the vertical purple for Law, or green for Code dashed line. For our upper baseline, we show the results of simultaneously fine-tuning the base model on a mixture of both tasks to obtain the multitask fine-tuning model (black star). This model sits at the upper right of the plane as it does not exhibit forgetting and performs well on both tasks. However, as stated before, in our continual learning setting where data streams in as a sequence of tasks, this is infeasible. As such, we implement a common continual learning method: rehearsal with a data buffer. Rehearsal is a common technique for mitigating forgetting in continual learning (Prabhu et al., 2023; Garg et al., 2024). It involves maintaining a buffer of past task data and interleaving it with new task data during fine-tuning (Robins, 1995), where the size of the buffer is a hyperparameter. In our baseline setup, each batch is constructed using streams that ensure it maintains x% of samples from the current task with 100-x% of samples randomly drawn from the buffer of past tasks, ensuring continual exposure to prior knowledge while prioritizing adaptation to new data. For simplicity, the stream randomly draws from the full training data of past tasks. We demonstrate its effectiveness by further training our initial model (dark blue circle, fine-tuned on Math) on a mixture of 90% (95%) task 2 data and 10% (5%) of Math data sampled randomly from the full Math dataset. The resulting continual learning (CL) with data buffer model (pink diamond in Figures 1, 2 and 11) effectively improves on the initial model on task 2 (higher Law performance, i.e. x-axis) while mitigating forgetting (maintains high Math performance i.e. y-axis). As we increase the proportion of Math data from 5% to 10%, we see higher performance on Math (Figure 2). However, a data buffer and has significant drawbacks: it requires storing data from all previous tasks, leading to rapidly increasing storage costs as the number of tasks and the size of the buffer grow. It also adds to the training cost, because we must continue to train on data from past tasks. Furthermore, maintaining a subset of past data can also threaten data privacy and security (Li et al., 2024). Finally, our domain fine-tuning scenario is set up for a variety of models, with baselines that include no intervention, multitask fine-tuning, and rehearsal with a data buffer.

## 4.2 SFA on Cross Domain Data

Next, we test the ability of SFA to retain past domain performance when learning a new domain by implementing it and comparing it to a variety of other merging techniques. Recall that in SFA, we take a model that has already been fine-tuned on Task A, and while fine-tuning on Task B, every $pT$ steps we average the weights with the final model after fine-tuning on Task A and continue fine-tuning on Task B. We evaluate SFA with varying averaging frequency $p$ during cross-domain sequential fine-tuning. Figures 1 and 11 show that as $p$ decreases, signifying more frequent averaging with the initial model, we observe stronger retention of past domain knowledge (orange curve). By adjusting the averaging frequency ($p$), we control the balance between past and new knowledge retention given that the model has progressed on the new task. This is evident, because as $p$ decreases, the performance on Math (y-axis) increases, indicating stronger retention of task 1. Furthermore, there is minimal loss to the potential learning of task 2 (Law or Code on the x-axis). Notably, when fine-tuning on Math followed by Law, a $p$ of 0.25 yields results comparable to

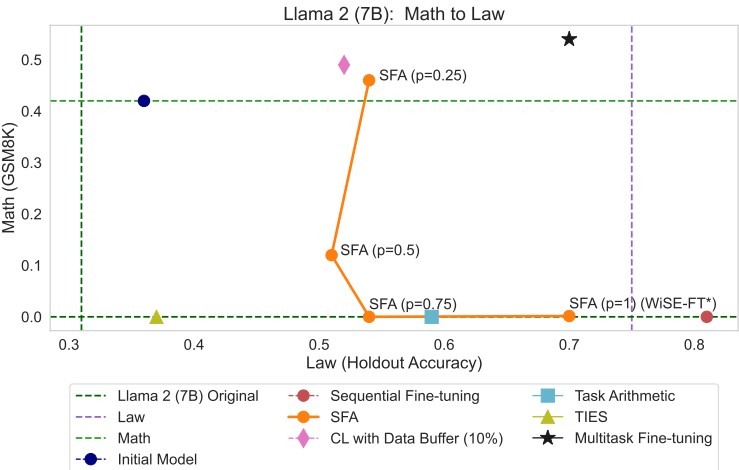

Figure 1: A comparison of Llama 2 (7B)'s performance on Math (y-axis) and Law (x-axis) using various fine-tuning and model merging techniques. The results are contained by dashed boundary boxes: the left and bottom lines represent the performance of a pretrained Llama 2 (7B) on Math and Law, whereas the right and top lines represent the performance of Llama 2 (7B) after fine-tuning on Law and Math respectively. A curve shows the performance of SFA with varying $p$, next to comparisons of continual learning with a data buffer, Task Arithmetic, and TIES. Finally, we also show an initial model (fine-tuned on math) and performance after sequentially fine-tuning it on Law.

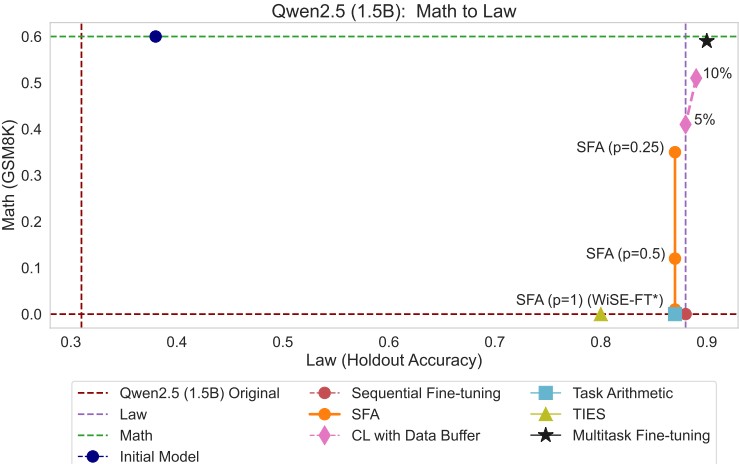

Figure 2: A comparison of Qwen2.5 (1.5B)'s performance on Math, Law using various fine-tuning and model merging techniques similar to Figure 1. On Math to Law, SFA $p = 0.25$ can be seen as having comparable performance to using a data buffer (5% past task), while outperforming Task Arithmetic, which resembles fine-tuning with no intervention and WiSE-FT in performance.

rehearsal (pink diamond), demonstrating that SFA can mitigate forgetting without the need for data buffers. Crucially, our method is able to achieve such performance without requiring a data buffer, but just two model checkpoints: the initial and current one throughout fine-tuning.

Additionally, in this sequential fine-tuning scenario, our method also outperforms other model merging methods. We implement Task Arithmetic (Ilharco et al., 2023) (blue square), TIES (Yadav et al., 2023) (green triangle), and WiSE-FT (Wortsman et al., 2022b), and show that our method achieves superior performance to all of these. In the Math-then-Law fine-tuning setting, we find that both of these methods,

Task Arithmetic and TIES, fail to retain Math performance completely, whereas SFA with a low enough $p$ is able to achieve performance on par with rehearsal. Our figure values for Pythia (2.8B) can be found in Table 2 (Math and Law), and Table 6 (Math and Code). Results for Llama 2 (7B) can be found in Table 4 (Math and Law), and Table 7 (Math and Code). Finally, results for Qwen2.5 (1.5B) can be found in Table 5 (Math and Law).

To see how our method scales as the number of domains increases, we also continue fine-tuning and applying SFA on our model for 3 domains (Figure 12). In these graphs, we take a high performing SFA model ($p$ of 0.25) on Math and Law, and Math and Code from Figure 11, and continue fine-tuning the model with SFA on the final domain (Code and Law respectively). We find that by using SFA (specifically adjusting $p$), we are able to maintain high performance on the previous 2 domains while also learning an additional domain. As such, SFA is a useful forgetting mitigation technique for continual learning given a sequence of domains. In both scenarios, Math-Code to Law, and Math-Law to Code, SFA (orange curve) outperforms Task Arithmetic, WiSE-FT, and sequential fine-tuning. In the case of Math-Code to Law, SFA with $p$ of 0.25 yields performance comparable to rehearsal (pink diamond). The figure results of Pythia (2.8B) fine-tuning on Math-Code to Law, and Math-Law to Code can be found in Table 8. As such, we show that in domain fine-tuning, SFA outperforms other merging techniques, while providing comparable performance to using a data buffer without storing any additional past domain data.

### 4.3 Mitigating Forgetting Across Sequences of Image Tasks

Building on SFA's success with dissimilar task pairs in the language domain, we next evaluate its performance on image classification tasks to provide a comprehensive empirical analysis across settings. Specifically, we demonstrate SFA's effectiveness in a continual learning scenario, where a ViT model is fine-tuned on a long sequence of image classification tasks drawn from both Food-101 and CIFAR-100 (Figure 3). We provide an upper baseline by simultaneously fine-tuning the initial model on all tasks to obtain a multitask fine-tuning model which performs well on average (black star). However, as previously stated, this is infeasible given a continual learning setting where task data appears sequentially. Next, we fine-tune without intervention (red bar) which results in low average accuracy due to the catastrophic forgetting of earlier tasks. To mitigate this forgetting, we implement 2 variations of Rehearsal with a data buffer: one that combines 5% past task data and 95% current task data, and another with 10% past task data and 90% current task data (pink dashed horizontal lines). These variations allow us to create a Rehearsal region of commonly used data buffer sizes to compare other methods to. As in the previous examples, we use the full training data of past tasks and randomly sample from it to ensure that 100-x% of the current training data is from past tasks. We also compare SFA to CLS-ER (Arani et al., 2022), which combines moving averages with a fixed memory capacity (ranging from 20-200 in our examples). In Food-101, SFA—either on its own or augmented with a fixed buffer matched in size—achieves performance comparable to CLS-ER configured with very small and medium buffers, respectively. On CIFAR-100, parity at larger capacities requires a larger fixed buffer for SFA; nevertheless, even in that regime SFA + FM remains less expensive, because storage costs are dominated by CLS-ER's additional model checkpoint requirement (Appendix A.11). Overall, SFA provides strong standalone performance and, when paired with a fixed buffer, constitutes a more cost-effective alternative to CLS-ER.

Similarly to Section 4.1, we next apply commonly used model merging methods to compare our original SFA to. Task Arithmetic (Ilharco et al., 2023) (blue bar), and WiSE-FT (Wortsman et al., 2022b) (orange bar). As we explain in Section 3, generalized WiSE-FT can be equivalent to SFA with $p = 1$, because it only involves merging the final trained and initial model. Finally, we compare these baselines to our method, SFA, which merges a partially-trained and an initial model, using varying averaging frequency $p$. SFA outperforms most other methods on both Food-101 and CIFAR-100, and performs comparably to using a reasonably sized data buffer (as well as to CLS-ER when combined with a fixed buffer). As expected, most methods except interestingly Task Arithmetic, outperform using no intervention. SFA with varying $p$ generally outperforms both using a smaller-sized data buffer and Task Arithmetic. Finally, SFA ($p = 0.98$ for Food-101, and $p = 0.96$ for CIFAR-100) where averaging occurs near the end and after training, achieves higher performance than WiSE-FT or SFA ($p = 1$) where averaging occurs only once at end. This suggests that averaging models during training is more effective than averaging only at the end of training, indicating an inherent difference in learning dynamics when an averaged model continues training on some task. However, the performance gap

between WiSE-FT and SFA is less severe than when tested against dissimilar language tasks in Section 4.2. This may suggest that the image classification tasks we evaluate on are more similar to each other than the sequential language tasks. As a result, infrequent averaging methods (e.g. WiSE-FT) can perform comparably to SFA due to the higher degree of shared knowledge across tasks.

In order to better understand why the optimal $p$ is so high in this scenario, we investigate the effects of averaging throughout this training cycle. We track the accuracy (y-axis) of both past and current tasks as a model fine-tunes on a sequence of tasks from Food-101 (Figure 4). We compare a model fine-tuning without intervention (top graph) to fine-tuning using SFA (p=0.98) (bottom graph). As is shown, without intervention, past task performance continues to decrease as new tasks are introduced. Meanwhile, SFA boosts past task performance when averaging occurs (performance spikes in graph). Furthermore, as can be seen, each task requires a substantial portion of training steps to raise its performance on the given task. This may explain why SFA ($p = 0.98$) outperforms SFA ($p < 0.98$), as averaging before substantial learning on the given task has taken place may hinder current task performance. We contrast this to Section 4.2, where averaging earlier on in training can be beneficial for more quickly-learned tasks. As such, while we show SFA's efficacy across a range of baselines in both language and image task fine-tuning, we find that it is especially effective on sequential language domains. As such, we recommend its implementation in both vision and language, with an emphasis on the latter due to the larger performance gaps.

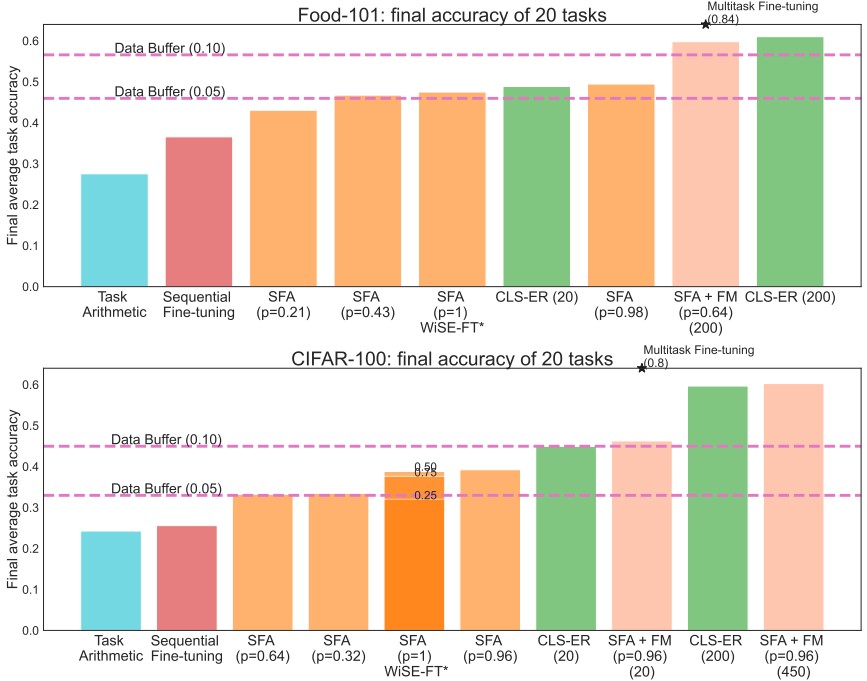

Figure 3: A comparison of ViT (base) fine-tuned on a sequence of 20 tasks from Food-101 (**top**) and CIFAR-100 (**bottom**) using various continual learning techniques. Across both datasets, using SFA with varying $p$ results in a high final average accuracy across all tasks (y-axis) comparable to using a data buffer. Furthermore, averaging during training ($p < 1$) achieves higher performance than only once at the end ($p = 1$). Finally, the combination of SFA and a fixed memory achieves performance comparable to CLS-ER, but is much cheaper, particularly as models scale up.

## 4.4 Averaging Weights

In order to better understand whether SFA's performance advantages are based on its continual averaging, we explore the impact of modifying weights solely at the final stage. Rather than using default averaging which weights each model equally, we vary the weight hyperparameter of each model and compare this to using

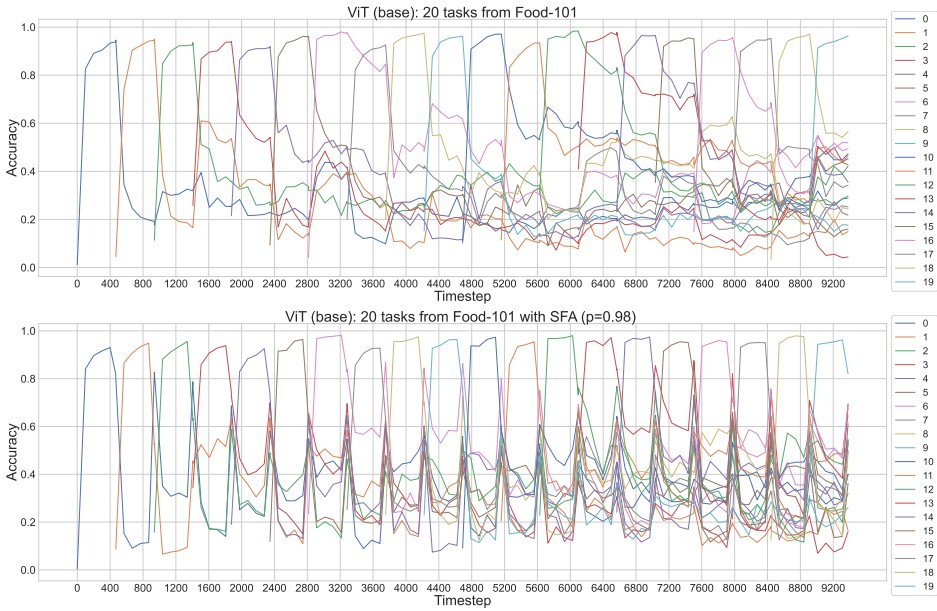

Figure 4: A comparison of sequentially fine-tuning ViT (base) on 20 tasks (Food-101) with (**bottom**) and without SFA (**top**). Each new task is introduced with a different colored curve across gradient timesteps (x-axis) resulting in changes to both current and past task accuracies (y-axis). The use of SFA can be seen to improve cumulative past task performance at averaging steps.

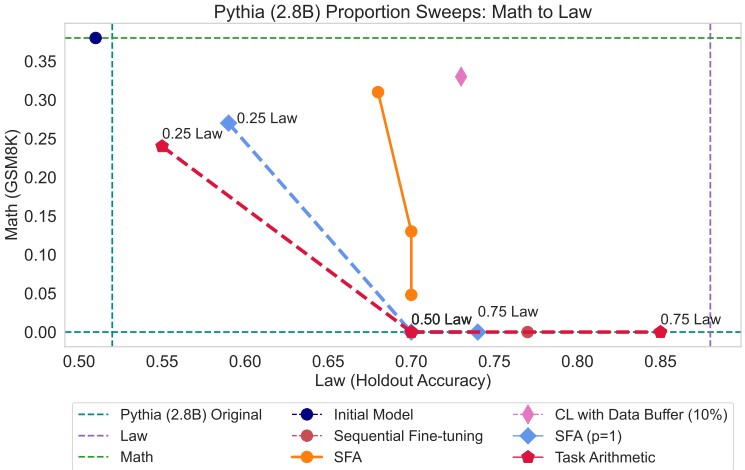

Figure 5: A comparison of varying the Task Arithmetic model weights, and $\beta$ on SFA ($p$=1), with SFA (varying $p, \beta = 0.5$) for Pythia (2.8B). We reproduce the results varying $p$ in SFA (orange curve) from Figure 11 and add 2 sweeps showing change in performance on Pythia (2.8B) when the weights for the current and past checkpoints are varied for SFA ($p = 1$) (dashed blue) and the domain-specific models are merged in Task Arithmetic (dashed red). Generally, SFA with $p < 1$ achieves highest performance, followed by SFA ($p = 1$) with varying weights, and lastly is Task Arithmetic with varying weights.

SFA. Our results underscore the importance of SFA's *continual* averaging approach for achieving optimal performance across multiple domains.

Recall that SFA combines parameters from the initial and current model during fine-tuning. We posit that the initial model represents expertise in past tasks/domains, while the current model embodies new task/domain

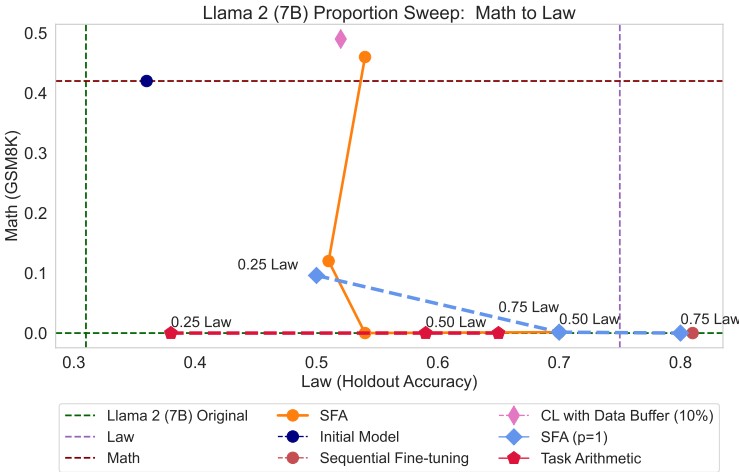

Figure 6: A comparison of varying the Task Arithmetic model weights, and $\beta$ on SFA ($p$=1), with SFA (varying $p$, $\beta = 0.5$) for Llama 2 (7B). We reproduce the results varying $p$ in SFA (orange curve) from Figure 1 and add 2 sweeps for the weights on the checkpoints and domain models of SFA ($p = 1$) and Task Arithmetic, similarly to Figure 5, to compare SFA with merging at different proportions. We see a similar outcome, where SFA with $p < 1$ generally achieves a better trade off in performance between Math and Law.

knowledge. Our default parameter weighting (0.50 for each) provides a balance. We explore if, instead of varying $p$, the frequency of averaging in SFA, we can get similar flexibility by first fine-tuning the model on a new task ($p = 1$, or WiSE-FT) and then averaging the final model with the previous task model using different relative weights (vary $\beta$). In Figures 5 and 6, we show that SFA with $p < 1$ and $\beta = 0.5$ (orange curve) performs the same if not better than a sweep of weighting parameter $\beta$ for SFA ($p = 1$, or WiSE-FT) (blue curve). Furthermore, for SFA ($p = 1$, or WiSE-FT) with $\beta \geq 0.50$, the trade off between Math and Law for both Pythia (2.8B) and Llama 2 (7B) is especially large, resulting in the complete failure to retain math. Likewise, for CIFAR-100 in Figure 3, we show that varying $\beta$ for SFA ($p = 1$, or WiSE-FT) is not as effective as SFA ($p = 0.96$), implying that averaging during fine-tuning, with additional fine-tuning afterwards offers additional performance benefits. This suggests that SFA's continual averaging during fine-tuning is key to its success in preserving cross-domain, and sequential task competence.

We extend this analysis to Task Arithmetic, another model merging technique. In Figures 5 and 6 we report the results sweeping over the weight values for averaging (red curve), and observe that Task Arithmetic, like SFA ($p = 1$, or WiSE-FT) with varying $\beta$, fails to achieve the cross-domain performance improvements that SFA demonstrates. Specifically, it also shows even worse combined performance on task 1 (Math, y-axis) and task 2 (Law, or Code, x-axis). Furthermore, in the Math-Law setting, for weights on Law $\geq 0.50$, it also fails to retain Math. As such, SFA $p < 1$ with $\beta = 0.50$ offers superior performance for cross domain fine-tuning on both tasks even when accounting for proportion sweeps. Therefore, we show that SFA's continual averaging approach is key to its performance benefits, because it still outperforms other merging techniques that only merge at the final stage even when averaging weights are varied.

## 5 SFA and L2-Regression

### 5.1 Intuition for Model Merging

In this section, we connect model merging to penalty-based regression methods to offer intuition for why merging is so effective. Specifically, there exist many penalty-based methods in continual learning where the forgetting of past tasks is mitigated by constraining training weights. The penalty is often used to prevent weights from straying from model weights that perform well on past tasks. Some methods include L1 and L2 penalty, as well as EWC (Kirkpatrick et al., 2017). Typically, these methods add a penalty to an existing

loss objective for every gradient step. This becomes computationally expensive as models scale, because for each gradient step, multiple copies of model weights have to be loaded in memory to calculate the penalty (e.g. the initial and currently training model), in addition to potential gradients. However, we can show, that SFA roughly approximates an existing penalty method, L2-regression, but is more feasible to implement. Consider, starting with $\theta_0$, the model trained on the previous task and $\theta_t$, the model currently being trained on the new task. Calculating the loss with an L2 penalty takes the form:

$$L(\theta_t) = L_{\text{task}}(\theta_t) + \frac{\lambda}{2}||\theta_t - \theta_0||^2. \tag{1}$$

Updating the model once using the gradient of this loss is equivalent to:

$$\theta_{t+1} = \theta_t - \eta(\nabla_{\theta_t} L_{\text{task}} + \lambda(\theta_t - \theta_0)). \tag{2}$$

This can be rewritten as:

$$\theta_{t+1} = (1 - \eta\lambda)\theta_t + (\eta\lambda)\theta_0 - \eta\nabla_{\theta_t} L_{\text{task}}. \tag{3}$$

Now we can compare this to a case of SFA where averaging occurs after each gradient step. Following the setup in Algorithm 1 given some $T$, for each gradient step, current model parameters are first updated using only task loss, before being averaged with the initial model:

$$\theta_{t+1}^* = \theta_t - \alpha\nabla_{\theta_t} L_{\text{task}}. \tag{4}$$

$$\theta_{t+1} = (1 - \beta)\theta_{t+1}^* + \beta(\theta_0). \tag{5}$$

We can combine these 2 steps to get the following form:

$$\theta_{t+1} = (1 - \beta)(\theta_t - \alpha\nabla_{\theta_t} L_{\text{task}}) + \beta(\theta_0). \tag{6}$$

This is equivalent to

$$\theta_{t+1} = (1 - \beta)\theta_t + (\beta)\theta_0 - \alpha\nabla_{\theta_t} L_{\text{task}}(1 - \beta). \tag{7}$$

As such, Equations 3 and 7 are equivalent if $\beta = \eta\lambda$ and $\alpha = \frac{\eta}{(1-\eta\lambda)}$. However, in practice, the benefit of SFA is that averaging occurs infrequently ($p$), rather than after every gradient step, which is computationally advantageous relative to L2-regression. As such, while SFA is not typically equivalent to L2-regression, the resemblance between Equations 3 and 7, suggests SFA can be understood as roughly approximating L2-regression. To further support this, we also offer an empirical example that shows the correlation between the L2 distance of the initial and training model in the context of our experimental setup (Appendix A.7), as well as suggest that SFA may have Bayesian motivation because of its similarity to L2-regression (Appendix A.6). In regards to other penalty methods, EWC can also be approximated as a model merging technique (Appendix A.8). We emphasize these connections to bridge model merging with classical continual learning, and offer intuition for why merging is effective at mitigating forgetting.

## 5.2 Performance Comparison of SFA and L2-regression

Given the similarity between SFA and L2-regression, it is reasonable to assume that they perform comparably. However, we find that SFA outperforms L2-regression, and EWC, thereby implying that infrequent averaging offers an advantage to imposing a penalty at each step. In Figure 7, we train a small, custom neural network on 2 sequential MNIST tasks (Task A and Task B) separated by label introduced in Moriarity (2020). Task A involves labelling the first 5 even numbers, whereas Task B labels the first 5 odd numbers. The blue dot refers to the model after training on Task A, whereas the red dot is additionally trained on Task B without intervention. As such, performance rapidly drops on Task A as the model optimizes for Task B. The solid orange curve refers to SFA where, in Figure 7 (top) we vary the averaging weight $\beta$ from Algorithm 1 and in Figure 7 (bottom) we vary the frequency of averaging in number of batches. As such, placing a higher $\beta$ or lower number of batches before averaging results in a model that performs better on Task A, and vice versa. The green dotted line shows L2-regression where $\lambda$ (weight on L2 penalty) varies, with a higher $\lambda$ performing better on Task A (and vice versa). Finally, an orange dotted line shows EWC with varying $\lambda$ (weight on EWC penalty) with a higher weight performing better on Task A (and vice versa). L2-regression outperforms EWC with a better trade off between performance on Task A and B. Interestingly, SFA outperforms both L2-regression and EWC when hyperparameters are optimized. As such, not only is SFA computationally much cheaper due to infrequent averaging, but it is also able to outperform imposing a penalty at every step.

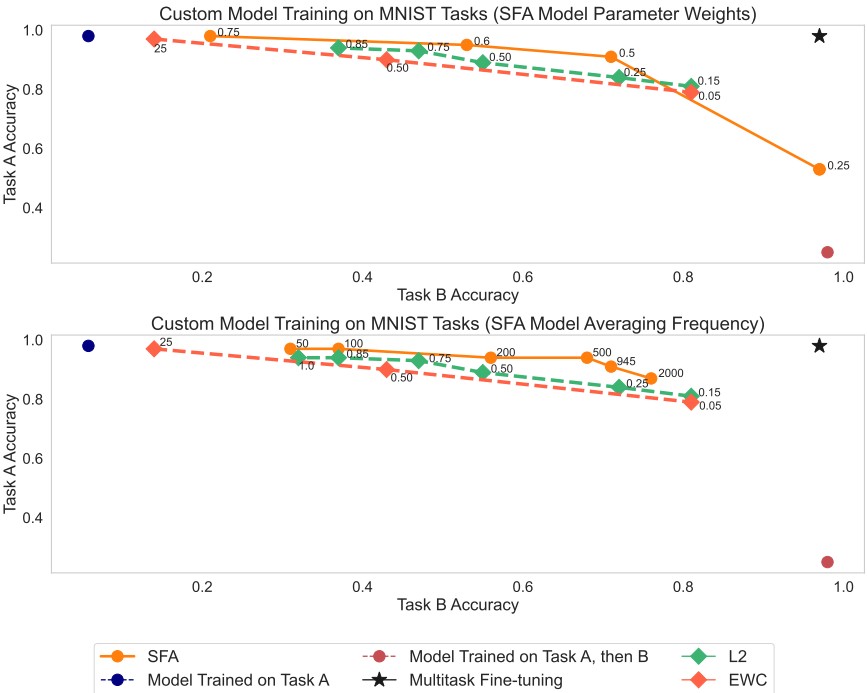

Figure 7: SFA compared against other continual learning methods, where the two tasks (Task A and B) were created by splitting MNIST by label. The accuracy after single-task training, sequential training, and multitask training is also shown. The lines for EWC and L2 are created by varying the coefficient corresponding to each method (and are the same for the top and bottom plots). **(Top)** visualizes SFA performance under varying $\beta$ coefficient, which determines how much weight is being placed on the initial model. **(Bottom)** visualizes SFA with varying averaging frequency in steps before averaging.

# 6 Conclusion

In this paper, we aim to mitigate the forgetting of past tasks in a continual learning setting. To this end, we introduce SFA, a novel approach for continual learning that merges a currently training model with a checkpoint from a previous task *during* training, rather than solely at the end. By relying on averaging frequency ($p$), we provide a mechanism to control how often this merging occurs, enabling a tunable balance between retaining prior knowledge and adapting to new tasks. Furthermore, SFA leverages previous model checkpoints as effective proxies for past data, requiring no additional memory or compute for rehearsal. Through extensive experiments, we show that SFA consistently outperforms post-hoc merging strategies, and achieves comparable performance to rehearsal with just a data buffer. Our evaluation spans two comprehensive settings: (1) fine-tuning large language models on sequential tasks with large domain shifts (e.g., Law, Math, Code), and (2) fine-tuning image classification models on long sequences of tasks. These settings test SFA's robustness under both large domain shifts and extended task streams. Together, our findings highlight the practical effectiveness of during-training averaging and position SFA as a simple, scalable, and data-efficient solution for continual learning.

## Broader Impact Statement

This paper presents work whose goal is to advance the field of Machine Learning. There are many potential societal consequences of our work, none which we feel must be specifically highlighted here.

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

# A Appendix

## A.1 Models

We fine-tune a combination of encoder-decoder and decoder only models. Specifically, we measure forgetting on T0_3B (3B) and T0pp (11B) (Sanh et al., 2021), two models already pretrained and fine-tuned on many tasks, when sequentially fine-tuning on instruction tasks (Appendix A.9). We also fine-tune Pythia (2.8B) (Biderman et al., 2023), Qwen2.5 (1.5B) (Yang et al., 2024) and Llama 2 (7B) (Touvron et al., 2023) on tasks from different domains (Math, Law, Code) to measure performance on sequential learning, in addition to a variety of merging techniques (Section 4.2). Finally, we fine-tune vit-base-patch16-224-in21k (vit, 2020) on image datasets.

Mainly, we use Composer (Team, 2021) for fine-tuning and evaluation. For additional evaluation metrics, we also use Language Model Evaluation Harness (Gao et al., 2023). Finally, we create some model merging baselines using mergekit (Goddard et al., 2024).

## A.2 Hyperparameters

Among our experiments, we typically first run a standard sweep on various hyperparameters and select the best performing ones, or record the full sweep to analyze our results with. For example, we run sweeps to determine averaging frequency $p$ (Figures 1 to 3, 7, 11 and 12), and the model merging weight hyperparameter of baselines including SFA ($p = 1$), Task Arithmetic, EWC, and L2 (Figures 5 to 7 and 13). Finally, when actually fine-tuning, we also initially run a task-specific learning rate sweep (typically from 1e-4 to 1e-6) to ensure robust results. Among our language domains of Math, Code and Law, we find that a learning rate of 1.0e-5 performs well across models and sequential task pairs (an exception being Qwen2.5 (1.5B) fine-tuning on Code for which 1.0e-6 is better). For our image classification tasks, we find that a learning rate of 5e-5 is optimal for both CIFAR-100 and Food-101. For future implementations of SFA, we recommend running similar hyperparameter sweeps, particularly for finding an optimal $p$ (i.e., randomly selecting 3-5 values for $p$ across 0-1) as this will allow for generalization to a wide range of tasks and data.

## A.3 Instruction Datasets

We use language generation tasks described in (Scialom et al., 2022) to measure forgetting. These tasks are based on pre-existing datasets that we also reference here: Text Simplification (Simpl) (Wiki-Auto (Jiang et al., 2020)), Inquisitive Question Generation (InqQG) (Eli5 (Fan et al., 2019)), Headline Generation with Constraint (HGen) (Gigaword (Graff et al., 2003; Rush et al., 2015)), Covid QA (CQA) (COVID-QA (Möller et al., 2020)), and Twitter Stylometry (TwSt) (Tweets Dataset (Bin Tareaf, 2017)).

Note: We retrieve the data for COVID-fact from (Scialom et al., 2022)'s existing codebase. We reference it using (Scialom et al., 2022) due to a lack of other citation in the paper.

## A.4 Image and Language Datasets

In order to measure and mitigate forgetting, we fine-tune our models on both a stream of image classification tasks, and 3 distinct language domains: Law, Math and Code.
In our classical continual learning setting, we construct a stream of 20 tasks from Food-101 (Bossard et al., 2014), as well as a stream of 20 tasks from CIFAR-100 (Krizhevsky, 2009). For Food-101, we construct our tasks by grouping 5-labels together for all labels except 100. For CIFAR-100, we group 5-labels together for all labels.
For each language domain, we fine-tune our model on a dataset featuring domain-specific knowledge, as well as unique instruction tasks. For Law, we combine CaseHOLD (Zheng et al., 2021), Terms of Service (ToS) (Lippi et al., 2019; tos, 2023), and Overruling (Zheng et al., 2021) to create a more general Law dataset. For Math, we use MetaMathQA (Yu et al., 2023), and for Code we use MagiCoder110k (Wei et al., 2023). We believe that required task knowledge across these 3 domains is distinct with minimal overlap. As such, we purposefully aim to test our models' ability to generalize across a wide range of knowledge to measure the validity of our method under maximal domain shifts.

### A.5 Evaluation Metrics

Table 1: Evaluation metrics for each task and domain used in our work.

| Task/Domain | Eval Metric |
|---|---|
| Food-101 | Food-101 holdout set |
| CIFAR-100 | CIFAR-100 holdout set |
| Text Simplification (Simpl) | Text Simplification (Simpl) holdout set |
| Inquisitive Question Generation (InqQG) | Inquisitive Question Generation (InqQG) holdout set |
| Twitter Stylometry (TwSt) | Twitter Stylometry (TwSt) holdout set |
| Headline Generation with Constraint (HGen) | Headline Generation with Constraint (HGen) holdout set |
| COVID-fact | COVID-fact holdout set |
| Covid QA (CQA) | Covid QA (CQA) holdout set |
| Law | CaseHOLD, ToS, Overruling holdout sets |
| Math | GSM8K (Cobbe et al., 2021) |
| Code | HumanEval (Chen et al., 2021) |

### A.6 Bayesian Interpretation

We use the well known point that L2-regression has a Bayesian Interpretation (bay, 2018) to motivate our method:

Assume that the prior distribution of the ideal model $\theta_t^*$ for a past and current task is Gaussian with mean the initial model, $\theta_t^* \sim N(\theta_0, \tau^2 I)$ for some $\tau$. Furthermore, assume that the distribution $y$ given input $X$, model weights $\theta_t^*$, and a function $f$ is Gaussian with mean the output of the function given $X, \theta_t$ : $y \sim N(f(X, \theta_t^*), \sigma^2 I)$ As such, the posterior of $\theta_t^*$ is:

$$p(\theta_t^*|y, X, f) \propto exp[\frac{-1}{2\sigma^2}(y - f(X, \theta_t^*))^T(y - f(X, \theta_t^*)) - \frac{-1}{2\tau^2}(\theta_t^* - \theta_0)^T(\theta_t^* - \theta_0)]. \tag{8}$$

We can compute the Maximum a Posteriori (MAP) for $\theta_t^*$:

$$\hat{\theta}_t^* = argmax_{\theta_t^*} exp[\frac{-1}{2\sigma^2}(y - f(X, \theta_t^*))^T(y - f(X, \theta_t^*)) - \frac{-1}{2\tau^2}(\theta_t^* - \theta_0)^T(\theta_t^* - \theta_0)]. \tag{9}$$

$$\hat{\theta}_t^* = argmin_{\theta_t^*}(y - f(X, \theta_t^*))^T(y - f(X, \theta_t^*)) + \frac{\sigma^2}{\tau^2}(\theta_t^* - \theta_0)^T(\theta_t^* - \theta_0). \tag{10}$$

$$\text{Set} \frac{\sigma^2}{\tau^2} = \lambda : \tag{11}$$

$$\hat{\theta}_t^* = argmin_{\theta_t^*}(y - f(X, \theta_t^*))^T(y - f(X, \theta_t^*)) + \lambda(\theta_t^* - \theta_0)^T(\theta_t^* - \theta_0). \tag{12}$$

As such, L2-regression tries to solve this Bayesian interpretation (Equation 11). As shown previously, SFA approximates L2-regression. This suggests that SFA may have a Bayesian motivation.

### A.7 L2 distance and SFA

To further explore this intuition of SFA and its relation to constraining parameter weights in the context of our experimental setup, we also show how accuracy and L2 distance are correlated when fine-tuning on different language domains. We use the setup described in Figure 11 where our model first fine-tunes on Math, then Law. As Figure 8 shows, when proportion of fine-tuning before averaging $p$ decreases on SFA (purple curve), the L2 distance to the initial Math model decreases, while the accuracy on Math increases. This is in direct contrast to sequential fine-tuning without intervention (black pentagon), because of its much higher L2 distance to the initial model. As such, $p$ directly relates to L2 distance, as well as performance on previous tasks, because averaging frequency constrains how much model parameters can change from their initial positions. The values for this figure can be found in Table 3.

### A.8 EWC Approximated by Model Merging

Consider fine-tuning a model with an EWC penalty (Kirkpatrick et al., 2017) where $\theta_0$ and $\theta_t$ are the weights of the initial and fine-tuning model respectively. $\eta$ is a hyperparameter, and $F_0$ is a diagonal matrix with the

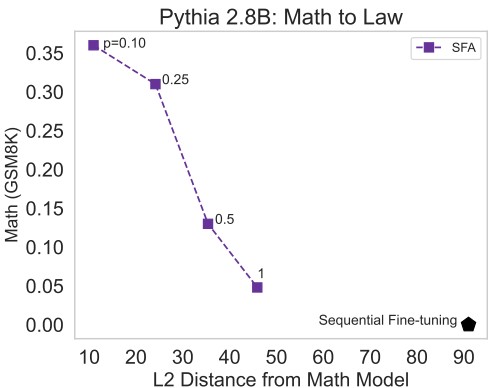

Figure 8: An analysis of the negative correlation between accuracy on Math and the $L2$ distance of the final model (fine-tuned on Math, then Law) from the original model (fine-tuned on Math only). The fine-tuning on Law is done using SFA with varying values of $p$ that determine the merging frequency. For reference we also mark sequential fine-tuning which leads to much higher L2 distance due to no merging, and accuracy just above that achieved with SFA merging once at the end of fine-tuning on law ($p = 1$).

initial model's Fisher information, and $\lambda = 1, j = 1, ..., |\theta|$.

$$L(\theta_t) = L_{\text{task}}(\theta_t) + \sum_j \frac{1}{2} F_0^{(j)}(\theta_t^{(j)} - \theta_0^{(j)})^2. \tag{13}$$

Assume that this loss update is split into 2 model updates. First, update model parameters using task loss on current weights, then update model parameters using EWC penalty:

$$\theta_{t+1}^* = \theta_t - \eta \Delta_{\theta_t} L_{\text{task}}. \tag{14}$$
$$\theta_{t+1} = (I - \eta F_0)\theta_{t+1}^* + \eta F_0 \theta_0. \tag{15}$$
$$\tag{16}$$

Thus, applying the EWC penalty can be understood as model merging weighted by the Fisher information of the initial model. This is reminiscent of Fisher model merging from Matena & Raffel (2022) where merging an initial and fine-tuning model has the form:

$$\theta^{*(j)} = \frac{\lambda_0 F_0^{(j)}\theta_0^{(j)} + \lambda_t F_t^{(j)}\theta_t^{(j)}}{\lambda_0 F_0^{(j)} + \lambda_t F_t^{(j)}}. \tag{17}$$

which can be rewritten as: $\tag{18}$

$$\theta^{*(j)} = \left(1 - \frac{\lambda_0 F_0^{(j)}}{\lambda_0 F_0^{(j)} + \lambda_t F_t^{(j)}}\right)\theta_t^{(j)} + \left(\frac{\lambda_0 F_0^{(j)}}{\lambda_0 F_0^{(j)} + \lambda_t F_t^{(j)}}\right)\theta_0^{(j)}. \tag{19}$$

Unlike the EWC approximation, this uses the Fisher information of both the initial and current model for merging.

## A.9   Forgetting under Sequential Fine-tuning

We confirm that fine-tuning on a sequence of different tasks leads to performance degradation on previously learned tasks. This forgetting phenomenon occurs across different task domains and for different model sizes. In this work, we focus on catastrophic forgetting of capabilities acquired during instruction fine-tuning instead of base pretrained model capabilities. This is because, as we will show, forgetting of skills learned during instruction fine-tuning can be quite severe and experiments at this scale are more feasible. We

fine-tune our models on a sequence of instruction, language generation datasets that test general knowledge to measure forgetting. Specifically, we use Scialom et al. (2022)'s: Text Simplification (Simpl), Inquisitive Question Generation (InqQG), Headline Generation with Constraint (HGen), COVID-fact, Covid QA (CQA), and Twitter Stylometry (TwSt). Many of these tasks incorporate existing datasets which we describe in Appendix A.3.

In our first experiments, we fine-tune the T0_3B (3B) and T0pp (11B) models (see Appendix A.1 for model descriptions) on the sequence of tasks described in Appendix A.4 while measuring forgetting on the first task. The results are shown in Figure 9. The model is first trained on Simpl which leads to a decrease in validation loss shown in blue. Subsequently, the model is trained on a sequence of other tasks; the decrease in validation loss on these tasks is shown in different colors. During this process, we continue to monitor the validation loss on Simpl, displayed in pink. As models fine-tune on new tasks, their performance on Simpl consistently declines as loss increases. This is true at both the 3B and 11B (Figure 9) model scales, indicating that merely scaling up parameter size does not help mitigate forgetting despite the increased capacity.

But how severe is this forgetting? We quantify this by comparing a model that was trained on and has then forgotten Simpl to a model that has never seen Simpl. In Figure 10, the pink line shows validation loss on Simpl for a model trained on a sequence of fine-tuning tasks starting with Simpl. As the model learns new tasks, its performance deteriorates. After 2000 steps, the sequentially fine-tuned model's loss on Simpl is the same order of magnitude as that of the multitask model trained on all tasks except Simpl. Thus if a model that has learned Simpl is finetuned on other tasks for as little as 2000 steps, its performance degrades to that of a model that has never seen Simpl. This indicates significant forgetting, as the model loses the ability to respond to tasks it previously was able to.

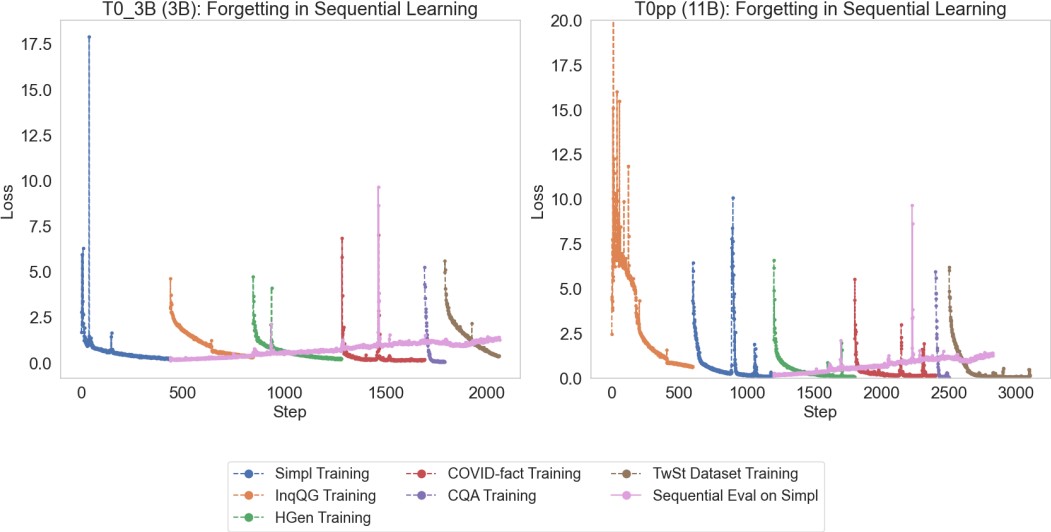

Figure 9: The fine-tuning of T0_3B (3B) and T0pp (11B) on a stream of language generation tasks. Training loss on each subsequent task decreases as the model learns it, while evaluation loss on Simpl continues to increase, indicating that forgetting is present.

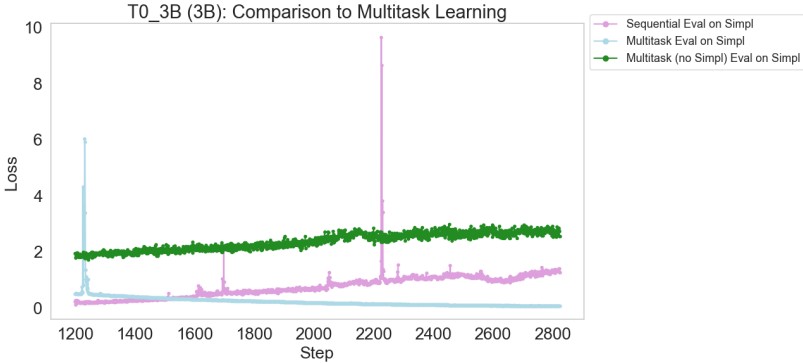

Figure 10: The Simpl loss curve of T0_3B (3B) from Figure 9 is compared to a multitask model training on all tasks, and a multitask model training on all tasks except Simpl. As T0_3B (3B) continues to fine-tune on each new task, the loss on Simpl becomes the same order of magnitude as that of a model that is never exposed to Simpl.

To summarize, we see a consistent trend of forgetting knowledge: as models are sequentially fine-tuned on new tasks, performance on past tasks drops resulting in lower evaluation metrics. This gets worse as more tasks are added and is not mitigated by model scale.

## A.10   Table Results of Domain Fine-tuning

Table 2: Results of Pythia (2.8B) models fine-tuning on Math and Law

| PYTHIA (2.8B) | CASE_HOLD | TOS | OVERRULING | GSM8K (0-SHOT) |
|---|---|---|---|---|
| PYTHIA (2.8B) ORIGINAL | 0.25 | 0.85 | 0.45 | 0 |
| METAMATHQA | 0.19 | 0.87 | 0.48 | 0.38 |
| LAW | 0.74 | 0.93 | 0.97 | 0 |
| METAMATHQA, LAW | 0.76 | 0.95 | 0.59 | 0 |
| METAMATHQA, LAW (P=1) | 0.74 | 0.88 | 0.49 | 0.048 |
| METAMATHQA, LAW (P=1, 0.75 LAW, 0.25 MATH) | 0.78 | 0.93 | 0.52 | 0 |
| METAMATHQA, LAW (P=1, 0.25 LAW, 0.75 MATH) | 0.42 | 0.87 | 0.49 | 0.27 |
| METAMATHQA, LAW (P=0.5) | 0.69 | 0.89 | 0.52 | 0.13 |
| METAMATHQA, LAW (P=0.25) | 0.67 | 0.87 | 0.49 | 0.31 |
| METAMATHQA, LAW (P=0.10) | 0.59 | 0.87 | 0.49 | 0.36 |
| TASK ARITHMETIC (0.5 LAW, 0.5 MATH) | 0.68 | 0.87 | 0.55 | 0 |
| TASK ARITHMETIC (0.75 LAW, 0.25 MATH) | 0.73 | 0.88 | 0.95 | 0 |
| TASK ARITHMETIC (0.25 LAW, 0.75 MATH) | 0.30 | 0.87 | 0.49 | 0.24 |
| MULTITASK | 0.76 | 0.87 | 0.58 | 0.40 |
| CONTINUAL LEARNING (DATA BUFFER 10%) | 0.72 | 0.93 | 0.54 | 0.33 |

Table 3: The L2 distance of Pythia (2.8B) models from previous checkpoints of models fine-tuning on Math and law

| PYTHIA (2.8B) | L2-DISTANCE |
|---|---|
| METAMATHQA, LAW - METAMATHQA | 90.99 |
| METAMATHQA, LAW (P=1) - METAMATHQA | 45.50 |
| METAMATHQA, LAW (P=0.5) - METAMATHQA | 35.38 |
| METAMATHQA, LAW (P=0.25) - METAMATHQA | 24.16 |
| METAMATHQA, LAW (P=0.10) - METAMATHQA | 10.94 |
| TASK ARITHMETIC (0.5 LAW, 0.5 MATH)- METAMATHQA | 101.77 |
| MULTITASK - METAMATHQA | 178.96 |
| CONTINUAL LEARNING (DATA BUFFER 10%) - METAMATHQA | 82.21 |

Table 4: Results of Llama 2 (7B) models fine-tuning on Math and law

| LLAMA 7B | CASE_HOLD | TOS | OVERRULING | GSM8K (0-SHOT) |
|---|---|---|---|---|
| LLAMA 2 (7B) ORIGINAL | 0.32 | 0.13 | 0.49 | 0 |
| METAMATHQA | 0.21 | 0.38 | 0.49 | 0.42 |
| LAW | 0.81 | 0.51 | 0.94 | 0 |
| METAMATHQA, LAW | 0.64, | 0.86 | 0.93 | 0 |
| METAMATHQA, LAW (P=1) | 0.61 | 0.59 | 0.90 | 0.0015 |
| METAMATHQA, LAW (P=1, 0.75 LAW, 0.25 MATH) | 0.64 | 0.83 | 0.94 | 0 |
| METAMATHQA, LAW (P=1, 0.25 LAW, 0.75 MATH) | 0.55 | 0.16 | 0.79 | 0.096 |
| METAMATHQA, LAW (P=0.75) | 0.53 | 0.13 | 0.97 | 0 |
| METAMATHQA, LAW (P=0.5) | 0.50 | 0.13 | 0.90 | 0.12 |
| METAMATHQA, LAW (P=0.25) | 0.53 | 0.13 | 0.95 | 0.46 |
| METAMATHQA, LAW (P=0.17) | 0.48 | 0.13 | 0.63 | 0.48 |
| TASK ARITHMETIC (0.5 LAW, 0.5 MATH) | 0.68 | 0.13 | 0.96 | 0 |
| TASK ARITHMETIC (0.75 LAW, 0.25 MATH) | 0.79 | 0.18 | 0.97 | 0 |
| TASK ARITHMETIC (0.25 LAW, 0.75 MATH) | 0.44 | 0.13 | 0.56 | 0 |
| TIES | 0.37 | 0.13 | 0.61 | 0.014 |
| MULTITASK | 0.86 | 0.27 | 0.97 | 0.54 |
| CONTINUAL LEARNING (DATA BUFFER 10%) | 0.46 | 0.13 | 0.96 | 0.49 |

Table 5: Results of Qwen2.5 (1.5B) models fine-tuning on Math and law

| QWEN2.5 1.5B | CASE_HOLD | TOS | OVERRULING | GSM8K (0-SHOT) |
|---|---|---|---|---|
| QWEN2.5 (1.5B) ORIGINAL | 0.47 | 0.28 | 0.90 | 0 |
| METAMATHQA | 0.26 | 0.19 | 0.69 | 0.60 |
| LAW | 0.77 | 0.91 | 0.95 | 0 |
| METAMATHQA, LAW | 0.78, | 0.92 | 0.94 | 0 |
| METAMATHQA, LAW (P=1) | 0.81 | 0.88 | 0.93 | 0.01 |
| METAMATHQA, LAW (P=0.5) | 0.77 | 0.89 | 0.94 | 0.12 |
| METAMATHQA, LAW (P=0.25) | 0.79 | 0.89 | 0.94 | 0.35 |
| TASK ARITHMETIC | 0.77 | 0.88 | 0.96 | 0 |
| TIES | 0.57 | 0.88 | 0.94 | 0 |
| MULTITASK | 0.83 | 0.90 | 0.96 | 0.59 |
| CONTINUAL LEARNING (DATA BUFFER 5%) | 0.78 | 0.93 | 0.93 | 0.41 |
| CONTINUAL LEARNING (DATA BUFFER 10%) | 0.79 | 0.93 | 0.95 | 0.51 |

Table 6: Results of Pythia (2.8B) models fine-tuning on Math and code

| PYTHIA (2.8B) | HUMANEVAL (5-SHOT) | GSM8K (0-SHOT) |
|---|---|---|
| ORIGINAL PYTHIA (2.8B) | 0.074 | 0 |
| METAMATHQA | 0.0 | 0.38 |
| MAGICODER-EVOL-INSTRUCT-110K | 0.15 | 0 |
| METAMATHQA, MAGICODER-EVOL-INSTRUCT-110K | 0.13 | 0.01 |
| METAMATHQA, MAGICODER-EVOL-INSTRUCT-110K (P=1) | 0.06 | 0.33 |
| METAMATHQA, MAGICODER-EVOL-INSTRUCT-110K (P=1, 0.3 MATH, 0.7 CODE) | 0.11 | 0.22 |
| METAMATHQA, MAGICODER-EVOL-INSTRUCT-110K (P=1, 0.6 MATH, 0.4 CODE) | 0.037 | 0.34 |
| METAMATHQA, MAGICODER-EVOL-INSTRUCT-110K (P=1, 0.7 MATH, 0.3 CODE) | 0.018 | 0.38 |
| METAMATHQA, MAGICODER-EVOL-INSTRUCT-110K (P=0.5) | 0.061 | 0.33 |
| METAMATHQA, MAGICODER-EVOL-INSTRUCT-110K (P=0.25) | 0.038 | 0.35 |
| TASK ARITHMETIC (0.5 CODE, 0.5 MATH) | 0.049 | 0.21 |
| TASK ARITHMETIC (0.75 CODE, 0.25 MATH) | 0.14 | 0 |
| TASK ARITHMETIC (0.25 CODE, 0.75 MATH) | 0 | 0.36 |
| MULTITASK | 0.13 | 0.35 |
| CONTINUAL LEARNING (DATA BUFFER 10%) | 0 | 0.32 |

Table 7: Results of Llama 2 (7B) models fine-tuning on Math and code

| LLAMA 2 (7B) | HUMANEVAL (5-SHOT) | GSM8K (0-SHOT) |
|---|---|---|
| LLAMA 2 (7B) ORIGINAL | 0.15 | 0 |
| METAMATHQA | 0 | 0.55 |
| MAGICODER-EVOL-INSTRUCT-110K | 0.35 | 0 |
| MAGICODER-EVOL-INSTRUCT-110K, METAMATHQA | 0.046 | 0.54 |
| MAGICODER-EVOL-INSTRUCT-110K, METAMATHQA (P=1) | 0.18 | 0.49 |
| MAGICODER-EVOL-INSTRUCT-110K, METAMATHQA (P=0.75) | 0.22 | 0.41 |
| MAGICODER-EVOL-INSTRUCT-110K, METAMATHQA (P=0.5) | 0.17 | 0.44 |
| MAGICODER-EVOL-INSTRUCT-110K, METAMATHQA (P=0.25) | 0.22 | 0.36 |
| TASK ARITHMETIC | 0.19 | 0.44 |
| TIES | 0.27 | 0.090 |
| MULTITASK | 0.09 | 0.40 |

Table 8: Results of Pythia (2.8B) models fine-tuning on Math, Law and Code for 2 orders

| PYTHIA (2.8B) | CASE_HOLD | TOS | OVERRULING |
|---|---|---|---|
| PYTHIA (2.8B) ORIGINAL | 0.25 | 0.85 | 0.45 |
| METAMATHQA | 0.19 | 0.87 | 0.48 |
| LAW | 0.74 | 0.93 | 0.97 |
| MAGICODER-EVOL-INSTRUCT-110K | 0.22 | 0.28 | 0.52 |
| METAMATHQA, LAW, MAGICODER-EVOL-INSTRUCT-110K | 0.30 | 0.87 | 0.51 |
| METAMATHQA, LAW, MAGICODER-EVOL-INSTRUCT-110K (P=1) | 0.50 | 0.88 | 0.59 |
| METAMATHQA, LAW, MAGICODER-EVOL-INSTRUCT-110K (P=0.5) | 0.55 | 0.88 | 0.57 |
| METAMATHQA, LAW, MAGICODER-EVOL-INSTRUCT-110K (P=0.25) | 0.57 | 0.88 | 0.67 |
| METAMATHQA, MAGICODER-EVOL-INSTRUCT-110K, LAW | 0.73 | 0.93 | 0.49 |
| METAMATHQA, MAGICODER-EVOL-INSTRUCT-110K, LAW (P=1) | 0.75 | 0.87 | 0.60 |
| METAMATHQA, MAGICODER-EVOL-INSTRUCT-110K, LAW (P=0.5) | 0.70 | 0.88 | 0.49 |
| METAMATHQA, MAGICODER-EVOL-INSTRUCT-110K, LAW (P=0.25) | 0.68 | 0.88 | 0.51 |
| TASK ARITHMETIC (0.33 MATH, 0.33 LAW, 0.33 CODE) | 0.63 | 0.87 | 0.88 |
| MULTITASK | 0.80 | 0.88 | 0.93 |
| CONTINUAL LEARNING (METAMATHQA, MAGICODER-EVOL-INSTRUCT-110K, LAW) (DATA BUFFER 10%) | 0.75 | 0.89 | 0.49 |
| CONTINUAL LEARNING (METAMATHQA, LAW ,MAGICODER-EVOL-INSTRUCT-110K) (DATA BUFFER 10%) | 0.69 | 0.89 | 0.56 |

| PYTHIA (2.8B) | GSM8K (0-SHOT) | HUMANEVAL (5-SHOT) |
|---|---|---|
| PYTHIA (2.8B) ORIGINAL | 0 | 0.0 |
| METAMATHQA | 0.38 | 0 |
| LAW | 0 | 0 |
| MAGICODER-EVOL-INSTRUCT-110K | 0 | 0.15 |
| METAMATHQA, LAW, MAGICODER-EVOL-INSTRUCT-110K | 0.01 | 0.14 |
| METAMATHQA, LAW, MAGICODER-EVOL-INSTRUCT-110K (P=1) | 0.34 | 0.068 |
| METAMATHQA, LAW, MAGICODER-EVOL-INSTRUCT-110K (P=0.5) | 0.37 | 0.051 |
| METAMATHQA, LAW, MAGICODER-EVOL-INSTRUCT-110K (P=0.25) | 0.39 | 0.055 |
| METAMATHQA, MAGICODER-EVOL-INSTRUCT-110K, LAW | 0.0 | 0.00 |
| METAMATHQA, MAGICODER-EVOL-INSTRUCT-110K, LAW (P=1) | 0.011 | 0 |
| METAMATHQA, MAGICODER-EVOL-INSTRUCT-110K, LAW (P=0.5) | 0.054 | 0 |
| METAMATHQA, MAGICODER-EVOL-INSTRUCT-110K, LAW (P=0.25) | 0.30 | 0.0012 |
| TASK ARITHMETIC (0.33 MATH, 0.33 LAW, 0.33 CODE) | 0 | 0.01 |
| MULTITASK | 0.38 | 0.22 |
| CONTINUAL LEARNING (METAMATHQA, MAGICODER-EVOL-INSTRUCT-110K, LAW) (DATA BUFFER 10%) | 0.30 | 0.029 |
| CONTINUAL LEARNING (METAMATHQA, LAW ,MAGICODER-EVOL-INSTRUCT-110K) (DATA BUFFER 10%) | 0.30 | 0.15 |

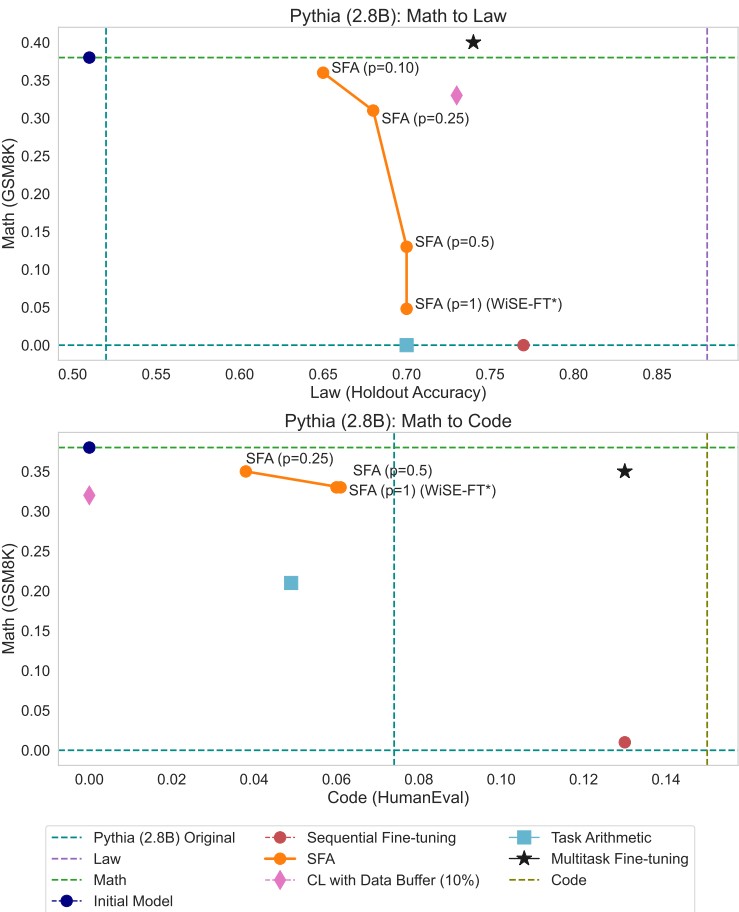

Figure 11: A comparison of Pythia (2.8B)'s performance on multiple domains (Math, Law and Math, Code) using various fine-tuning and model merging techniques similar to Figure 1. On Math to Law, SFA $p = 0.25$ can be seen as having comparable performance to using a data buffer, while outperforming Task Arithmetic. Likewise, in Math to Code, SFA with varying $p$ outperform using a data buffer and Task Arithmetic.

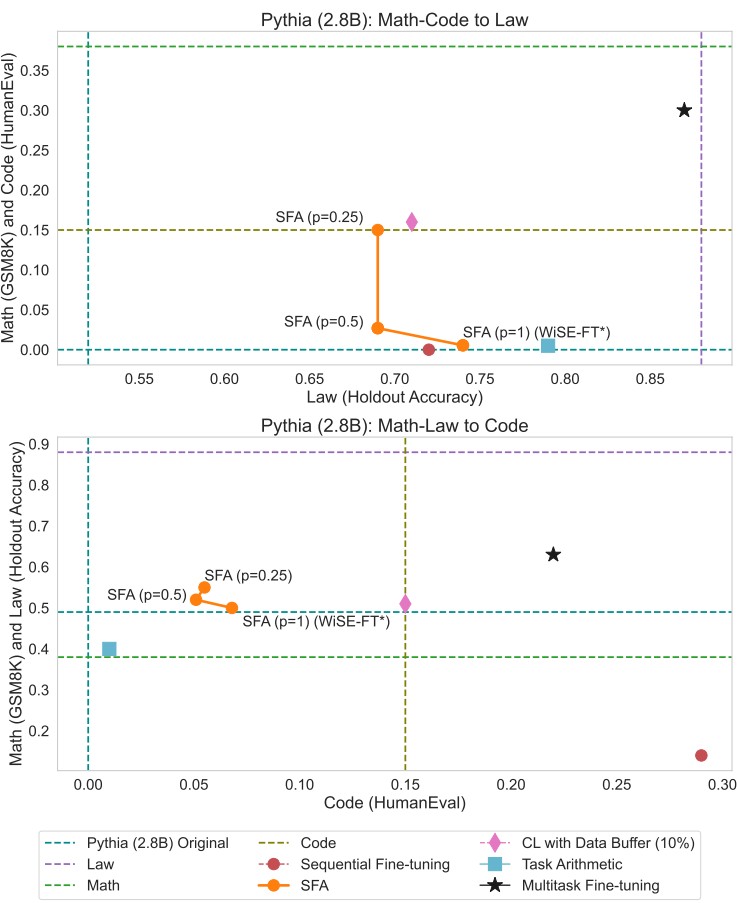

Figure 12: A comparison of Pythia (2.8B)'s performance when training on more than 2 domains (e.g. Math-Law and Code, Math-Code and Law) using various fine-tuning and model merging techniques similar to Figure 11. On Math-Code to Law, SFA $p = 0.25$ can be seen as having comparable performance to using a data buffer, while outperforming Task Arithmetic. While, SFA with varying $p$ on Math-Law to Code outperforms Task Arithmetic, but performs worse than using a data buffer.

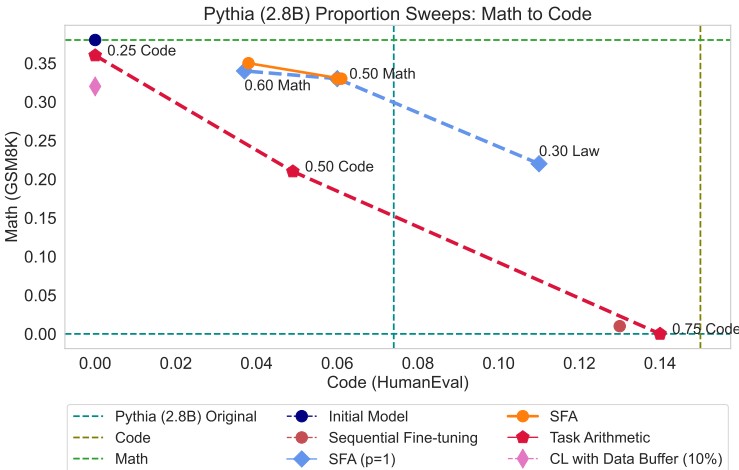

Figure 13: A comparison of varying the Task Arithmetic model weights, and $\beta$ on SFA ($p$=1), with SFA (varying $p, \beta = 0.5$) for Pythia (2.8B) on math then code, similar to Figure 5. Results are consistent as well where SFA with $p < 1$ achieves highest performance, followed by SFA ($p = 1$) with varying weights, and lastly is Task Arithmetic with varying weights.

### A.11 Comparing Storage Overhead

We conduct a straightforward memory usage comparison to illustrate how our method can substantially reduce storage and memory requirements in continual learning, relative to approaches that rely on a data buffer. Specifically, we consider a sequence of image classification tasks and fine-tune a ViT-Base model on two settings: (1) a sequence of CIFAR-100–sized tasks and (2) a sequence of ImageNet-1k–sized tasks (Russakovsky et al., 2015). We measure the number of bytes needed to store each method's additional requirements on disk—namely, the two sets of model parameters maintained for SFA's averaging, and the data buffer required for each gradient step. Furthermore, we also assume that 10% of each past task's data is added to the given buffer following training on that task with no fixed capacity. Given this, we measure the number of bytes using the following equations (assuming Uncompressed 8-bit storage for images, FP16 for model parameters, and standard preprocessing):
SFA $\approx 2$(model parameters)(bytes per param) $\approx 2(86e6)(2) \approx 0.34$GB

Data Buffer (10% added) $\approx$(0.1N)(H)(W)(C)(bytes per element)(number of past tasks)
CIFAR-100-sized Data Buffer (10% added) $\approx$(5000)(32)(32)(3)(1)(number of past tasks)
ImageNet-1k-sized Data Buffer (10% added) $\approx$(128117)(224)(224)(3)(1)(number of past tasks)

Multitask Fine-tuning $\approx$(N)(H)(W)(C)(bytes per element)(number of total tasks)
CIFAR-100-sized Multitask Fine-tuning $\approx$(50000)(32)(32)(3)(1)(50) $\approx 7.68 GB$
ImageNet-1k-sized Multitask Fine-tuning $\approx$(1281170)(224)(224)(3)(1)(50) $\approx 9,642.60 GB$

As shown in Figure 14, the storage costs of data buffer–based methods increase linearly with the number of tasks. For smaller datasets, the cost of maintaining a data buffer quickly exceeds that of storing two model checkpoints. This disparity becomes even more pronounced as dataset size increases, with SFA offering an immediate storage advantage over data buffer approaches. While hyperparameters such as data buffer implementation (e.g. fixed capacity) and size can be optimized (i.e., $\leq 10\%$ added past task data), the buffer will generally still increase with task sequence length up to a point, while the cost of SFA remains constant.
Beyond storage considerations, larger data buffers can also lead to longer training times depending on implementation. As past is combined with current task data during training, increased computational cost may be necessary to retain full past knowledge. In contrast, SFA maintains constant storage, as well as stable

GPU memory usage and GPU–CPU transfer latency, regardless of the number of past tasks. Consequently, when the number of future tasks is unknown and potentially large, SFA is generally more cost-effective to execute.

We also offer a cost comparison of our method to CLS-ER. In contrast to only using a data-buffer, the storage costs of CLS-ER are dominated by additional model storage in our examples, particularly for larger models. We measure the number of bytes similar to previously, but now for scaling models:

SFA $\approx$ 2(model parameters)(bytes per param) $\approx$ 2(P)(2)

SFA + Fixed Memory (Memory capacity 200, ImageNet-1k-sized tasks) $\approx$ 2(model parameters)(bytes per param) + (100)(H)(W)(C)(bytes per element)$\approx$ 2(P)(2) + (200)(224)(224)(3)(1)

CLS-ER (Memory capacity 50, ImageNet-1k-sized tasks) $\approx$ 3(model parameters)(bytes per param) + (50)(H)(W)(C)(bytes per element) $\approx$ 3(P)(2) + (50)(224)(224)(3)(1)

As illustrated in Figure 15, the storage overhead of CLS-ER rises sharply with model size. As models scale, the storage gap between CLS-ER and SFA widens, making CLS-ER less practical in settings where large models are commonplace. This is true even if SFA is combined with a much larger fixed memory capacity than CLS-ER (SFA + FM). CLS-ER also increases training cost: each example requires additional forward passes (for the semantic memories), and these accumulate into substantial time and resource demands. Taken together—storage and execution—SFA offers a more economical alternative for mitigating forgetting in continual learning.

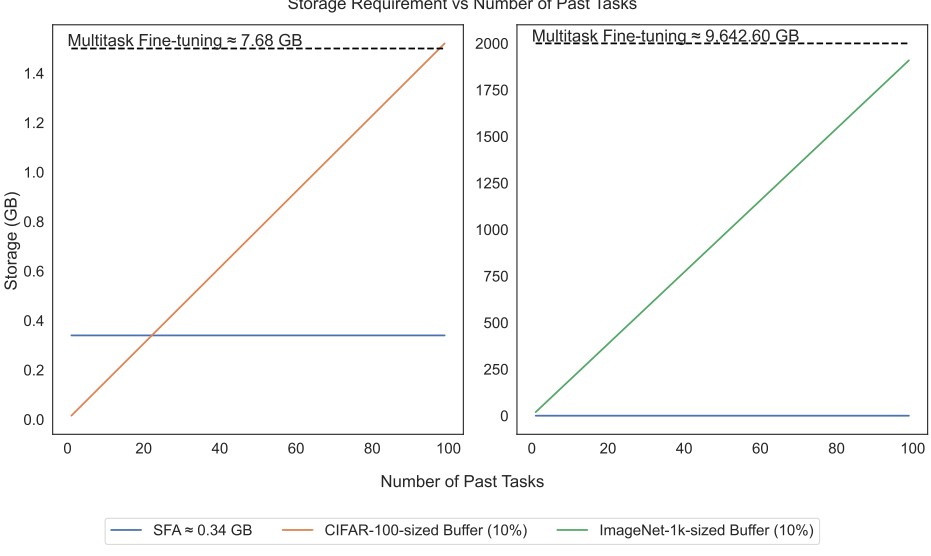

Figure 14: A comparison of storage when implementing SFA and using a data buffer during sequential task fine-tuning on smaller CIFAR-100-sized tasks (**left**) and larger ImageNet-1k-sized tasks (**right**). Using SFA is ultimately cheaper for longer sequences of smaller tasks, and is immediately cheaper when training on a larger sized image dataset.

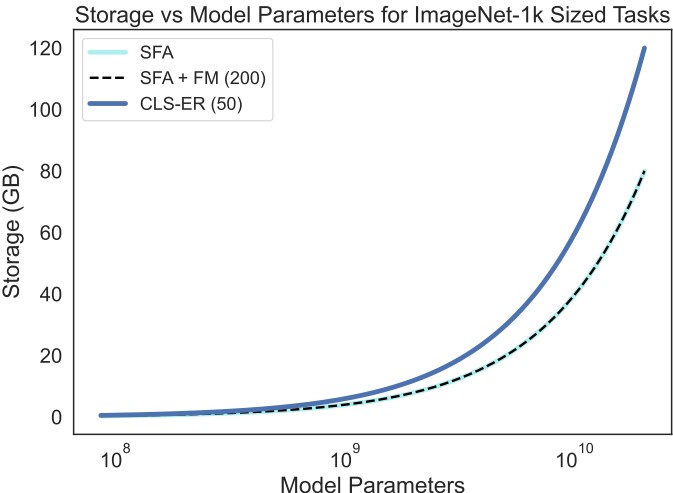

Figure 15: A comparison of storage when implementing SFA and using CLS-ER with scaling model size.

### A.12 SFA Without Task Boundaries

In CL, explicit task boundaries may not exist or may be unknown during training. While our work primarily focuses on the former, we also offer a solution to address the latter case by modifying SFA for training over a total of $T$ steps, which may span multiple tasks. Following from Algorithm 1, the parameters of the current model are periodically averaged with an earlier checkpoint every $C$ steps, after which training resumes from this averaged model. In parallel, the reference ('initial') model is updated every $A$ steps, allowing it to serve as a dynamic anchor that is not restricted by task boundaries. Both $C$ and $A$ are tunable hyperparameters, providing flexibility to adapt the method to a variety of CL scenarios.

---

**Algorithm 2** Sequential Fine-tuning with Averaging (Without Task Boundaries) (SFA-WTB)

---

$\quad$ **Input:** $\theta_0, C, A, \beta, T$
$\quad$ **for** $t$ in $0, ..., T-1$
$\quad\quad$ $\theta_{t+1}^* = \theta_t - \alpha \nabla_{\theta_t} L_t$
$\quad\quad$ **if** $(t+1) \mod C = 0$ **then**
$\quad\quad\quad$ $\theta_{t+1} = (\beta)\theta_0 + (1-\beta)\theta_{t+1}^*$
$\quad\quad$ **else**
$\quad\quad\quad$ $\theta_{t+1} = \theta_{t+1}^*$
$\quad\quad$ **if** $(t+1) \mod A = 0$ **then**
$\quad\quad\quad$ $\theta_0 = \theta_{t+1}$

---

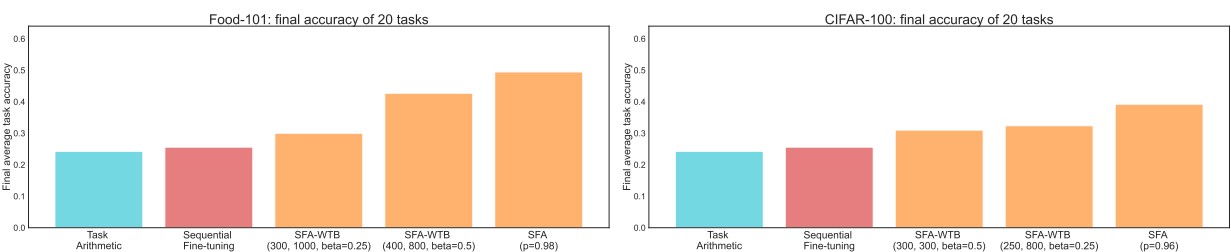

Figure 16: A comparison of SFA with and without using task boundaries to control model averaging and updating frequencies on a sequence of 20 tasks from Food-101 (**left**) and CIFAR-100 (**right**). While SFA is still effective without access to task boundaries, their incorporation enhances performance.

We next evaluate our updated method in the same vision scenarios described in Section 4.3. Specifically, we perform a hyperparameter sweep over $C$, $A$, and $\beta$ when training on a sequence of 20 tasks from Food-101 and CIFAR-100 (Figure 16). Our results show that SFA-WTB effectively mitigates forgetting in continual learning without requiring task boundary information. Nevertheless, the original SFA, which leverages explicit task boundaries, remains more effective at preserving performance. This is consistent with the underlying intuition: by treating the initial model as a proxy for past task knowledge, SFA benefits most when the checkpoint corresponds to the completion of a task, since it then contains the most accurate representation of prior knowledge. In contrast, SFA-WTB, lacking access to boundaries, may merge with checkpoints that are less well optimized for earlier tasks. Despite this limitation, SFA-WTB provides a low-cost and practical solution for mitigating forgetting in settings where task boundaries are unknown or unavailable.

