# OpenReview forum: "Soup to go: mitigating forgetting during continual learning with model averaging"
_TMLR — Rejected by TMLR_

### Review · Reviewer_5j8T · 2025-08-10

**Summary Of Contributions:**

This paper proposes a continuous learning method that can avoid forgetting previous knowledge. The method, SFA, utilize model averaging to mitigate forgetting. During the finetuning on a new task, the model checkpoint from learning the previous task is merged periodically. The paper shows the analysis of the connection of SFA to L2 penalty method used in classical continual learning algorithm.

**Audience:**

Yes

**Broader Impact Concerns:**

No concerns on broader impact

**Claims And Evidence:**

Yes

**Requested Changes:**

1. Please include a comparison on the memory overhead of storing checkpoint vs. storing previous data in finetuning
2. Please conduct a sequential finetuning experiment with multiple tasks on language model
3. Please report the final accuracy, average accuracy, and best accuracy of different tasks as tables instead of/besides figures for improved clarity in comparison

**Strengths And Weaknesses:**

## Strength
1. The proposed method is straightforward and easy to implement.
2. Both empirical and theoretical analysis are provided to show the effectiveness of the proposed method.

## Weakness
1. Novelty-wise, the difference between the proposed method and WiSE-FT seems to be incremental. As discussed in the paper, WiSE-FT appears to be a special case p=1 of the proposed method. In cases like the image tasks in Fig. 3, WiSE-FT appears as good as the proposed method. To this end, a clear discussion is needed to analyze when the proposed method can be helpful compared to WiSE-FT.
2. Results in Fig. 1-6 demonstrates that the choice of p has a significant impact on the performance of the proposed method. Its worth discussing how to find a proper p when facing a new sequence of tasks. It is especially interesting to consider if the choice of p may change given some different property of neighboring tasks or the sequence order of the tasks.
3. All results are represented as figures in the main paper, each including multiple different colors and shapes that are hard to distinguish and understand. This is especially the case for Fig. 4 where it's almost impossible to identify and compare the performance of a specific task. To this end, having some tables listing important statistics may better illustrate the effectiveness of the proposed method.
4. One key point argued by the paper compred to data buffer-based method is the reduced memory. However, for LLMs, model size may be significantly larger than the size of data used for finetuning in each task. Similarly, the ViT B model used for the vision task has 86M parameters while the full CIFAR-100 training set is only 5M in total. To this end, the proposed method may not be more efficient in memory compared to data buffer-based methods.

---

> ### Author Response · Authors · 2025-08-26
>
> We thank the reviewer for their comments and feedback. We address the listed Weaknesses and Requested Changes in order:
>
> Weaknesses:
>
> 1) We thank the reviewer for pointing this out, and we have added a more direct recommendation for when to use SFA.
>
> While SFA builds on existing merging ideas, it uses p to actively control the forgetting-plasticity trade-off during fine-tuning. To this end, performance is non-incremental where domain shifts are large (e.g., language tasks), precisely where end-of-task merging (p=1, as in, e.g., WiSE-FT) often fails, while SFA (p<1) succeeds (Figure 2, SFA (p=0.25) retains 0.35 on GSM8K while WiSE-FT is only able to retain 0.01). As such, SFA generalizes WiSE-FT while still consistently outperforming WiSE-FT among both image and language tasks, with the greatest gap in language.
> Furthermore, SFA is easy to implement for users already familiar with model merging, as it may only require a simple change to existing code bases for performance improvements.  As such, our work can be viewed as a generalization to WiSE-FT that consistently outperforms the former, with a large margin among language tasks.
>
> 2) Thank you, we have added additional clarification to emphasize our recommendation for finding p based on our results.
> In summary, we treat p as a standard hyperparameter that we perform a sweep on as is commonly done, and our graphs demonstrate this (e.g. Figure 1). This is what we currently promote as our recommendation for finding p given unique use cases and datasets. Furthermore, it appears that among language tasks, p=0.25 seems to be a generally good value. We have also experimented with sequence order of the tasks and found that best performing p values appear consistent.
>
> We agree that future work which focuses on efficiently selecting p would be very useful. Perhaps, given the range of unique tasks and data, the selection of p may be addressed through additionally training a model to infer optimal p range given tasks and current state, and sequential neighboring task properties as suggested.
>
> 3) We actually do provide tables for our main paper figures (See Tables 2-8 in the Appendix). We will add a table for specifically Figure 4 as well.
>
> 4) Thank you for this really interesting point. We have added a buffer to model storage comparison in the paper (Appendix section Comparing storage overhead) to address this and to show how SFA can require significantly less storage or computational cost compared to using a buffer.
> In summary, this comparison is focused on: a) task sequence length and b) the increased computational costs of retraining.
> a) As streams of tasks get longer, the data buffer generally grows to store additional tasks. As such, total storage grows with the number of past tasks (potentially to a fixed capacity). For both smaller datasets past a certain task threshold, or immediately for larger datasets, buffer storage exceeds SFA, which only relies on a constant amount of storage for 2 model weights. We provide a comparison using both ViT and CIFAR-100 and ImageNet-1k to show this relation.
> b) Furthermore, as data buffers grow with more past tasks, actual training becomes costlier, because of data iteration. As such, computationally, data buffers can be significantly more inefficient further along a task sequence without optimization techniques. On the other hand, SFA maintains constant storage and training costs, because it performs the same intermediate step: combining the weights of 2 model checkpoints.
>
> Requested Changes:
> 1) Thank you for this suggestion, we have added this to the paper.
> 2) We actually have already conducted such experiments (e.g. Figure 12 and Table 8).
> 3) We actually have tables with our main figure accuracies (See Tables 2-8) and will add a table for specifically Figure 4 as well.

---

### Review · Reviewer_gg1e · 2025-08-16

**Summary Of Contributions:**

The paper investigates whether model averaging, typically performed once at the end of training, can be applied multiple times during training to improve continual learning performance. The authors propose a simple rehearsal-free method where partially trained models are averaged at fixed checkpoints, aiming to reduce catastrophic forgetting while retaining plasticity. The method is evaluated across image datasets and LLMS. The paper also draws connections to weight regularization approaches, such as L2 regression, highlighting how model averaging can approximate L2 regularization.

**Audience:**

Yes

**Claims And Evidence:**

No

**Requested Changes:**

- Critical: Reconsider the novelty claims by discussing prior work on continual weight averaging in continual learning, particularly CLSER. .

- Provide details of rehearsal based approach used and consider well estblished rehearsal based methods. While the reviewer agrees that the paper is focusing on rehearsal free method, the claim that it can perform on par with rehearsal needs further investigation.

**Strengths And Weaknesses:**

Strengths
- Simplicity: The proposed method is lightweight and intuitive.
- Rehearsal-free focus: Avoids the use of memory buffers, making it appealing for scenarios with strict memory constraints.
- Insightful comparison: The connection between checkpoint averaging and L2 regularization provides useful theoretical grounding.
- Cross-domain experiments: Application to both vision and LLMs broadens the scope of evaluation.

Weakness:

My main concern with the paper and what constitutes as the main weakness is the claim and the weakness of emprical evaluation.

Author's set out to answer the following question:

> Why should model averaging occur only once at the end of training? Could averaging partially trained models help mitigate forgetting
while simultaneously improving performance on new tasks through additional training?

And they claim that they are the first to explore in-training averaging frequency on model performance. However, well established prior work such as CLSER [1] already investigates weight averaging at different frequencies during training of the new task. This omission weakens the positioning. Additionally CLSER does not make assumption about having knowledge of task boundaries which makes it suitable for general continual learning where the task boundaries are not necessarily distinct. The authors need to reposition their paper in light of the afforementioned paper.

While there are some differences, the authors use a fixed checkpoint which represents the optimal weights of the previous task, CLSER contnually aggregates the new weights as the model learns a new task, it does show the benefits of in-training averaging and insights into the main questions authors set out to answer. CLSER also have shown the effect of different averaging frequencies which the authors claim as another novelty of their approach.

Additionally, the comparison to rehearsal based approaches is weak, missing critical details as to how the  memory buffer is updated, which sampling technique is used and no stronger or recent rehearsal baseline is used.

Overall,
- Novelty claims are overstated: The authors claim to be the first to explore in-training averaging frequency, but prior work such as CLSER already investigates weight averaging at different frequencies during continual learning.

- Insufficient empirical evaluation: The baselines considered are limited. The paper lacks stronger and recent rehearsal-based baselines and does not describe buffer update strategies or sampling techniques in detail.

- Over-reliance on fixed checkpoint assumption: The method depends on identifying “optimal” weights of the previous task, while CLSER and other baselines demonstrate continual aggregation without assuming task boundaries.

- Presentation issues: Results and discussions are not always clearly presented, making it harder for readers to follow the key findings and implications.


[1] Arani, Elahe, Fahad Sarfraz, and Bahram Zonooz. "Learning Fast, Learning Slow: A General Continual Learning Method based on Complementary Learning System." International Conference on Learning Representations.

---

> ### Author Response · Authors · 2025-08-26
>
> We thank the reviewer for their review and address the weaknesses and requested changes in order:
>
> 1) Thank you so much for pointing us to this work. We have updated the paper’s novelty claims accordingly given CLSER’s earlier use of moving averages during training, as well as added: CLSER as a baseline (Section 4), a comparison of CLSER and SFA (Sections 1 and 2), and a storage comparison (Appendix: Comparing Storage Overhead).
>
> 2) We have implemented CLSER as an additional baseline (Section 4). Generally we find that SFA as a standalone method, or combined with a similarly sized fixed memory (such as in CLSER) performs comparably to CLSER. In the case of CIFAR-100, we find that as CLSER’s memory capacity grows, SFA must use a larger fixed memory to attain comparable performance to CLSER, however these added storage costs still allow SFA to be more cost effective than CLSER because of its additional model checkpoint (Appendix: Comparing Storage Overhead). Furthermore, during runtime, CLSER requires additional forward passes (for the semantic memories), and these accumulate into larger resource demands.
>
> As such, while SFA itself is a buffer-free method, it is comparable in performance to CLSER when combined with a buffer, while still being more cost effective.
>
> Furthermore, we have added details about our buffer method (Section 4). To summarize, for each new task, we randomly select from full past task training data such that x% of the training data is new data, and 100-x% is randomly selected past task data from all past tasks.
>
> 3) We are currently adding experiments that do not rely on the checkpoint assumption. In particular, we are adding a variant of SFA that performs parameter averaging and model updates at separate fixed intervals. This allows for direct comparison with other baselines that do not assume task boundaries.
>
> 4) We have tables in addition to Figures for our results (Tables 2-8) and would be happy to fix any additional figures/tables that are not clear.
>
> Requested Changes:
>
> 1) We have updated the paper to reconsider our previous novelty claims, as well as, have implemented CLSER as an additional baseline that we compare to SFA both in terms of performance and storage overhead.
> 2) We have provided additional information about our rehearsal based approach and have implemented CLSER as an additional baseline.

---

> ### Comment · Reviewer_gg1e · 2025-09-07
> **Reviewer'sreponse to Author's**
>
> Thank you for making the stated changes, it improves the readability of the paper and provides more context for readers. The addition of boundary free version, if and when available, would improve the applicability of the approach significantly.
>
> A general remark, please color your changes with a different color e..g blue so that the reviewers can know which parts of the revision to go though, as is, its not possible to review the changes without using a diff tool or reading though the entire text.
>
> Regarding, CLS-ER, the reviewer appreciates the effort of reviewers in addressing the raised issue. However, the reviewer believes the authors misunderstood the main issue raised, CLS-ER was mentioned as one of the prior papers that already studies in training averaging, since then there has been several methods that leverages in-training model averaging and addresses the storage issues of CLS-ER while improving the empirical performance, here are some of the recent papers which the reviewers believes the authors should be aware of since they are highly relevant to their study, and model averaging has been explored both with and without replay buffer and having only a single averaged model. In general, this papers lacks a thorough study of the relavant literature and positioning of author's contribution in comparision to earlier studies (not exhausitive, there are several more even from the same lab).
>
> [1] Döbler, Mario, Robert A. Marsden, and Bin Yang. "Robust mean teacher for continual and gradual test-time adaptation." Proceedings of the IEEE/CVF Conference on Computer Vision and Pattern Recognition. 2023.
>
> [2] Soutif–Cormerais, Albin, Antonio Carta, and Joost Van de Weijer. "Improving online continual learning performance and stability with temporal ensembles." Conference on Lifelong Learning Agents. PMLR, 2023.
>
> [3] Sarfraz, Fahad, Elahe Arani, and Bahram Zonooz. "Error Sensitivity Modulation based Experience Replay: Mitigating Abrupt Representation Drift in Continual Learning." The Eleventh International Conference on Learning Representations.
>
> [4] Bhat, Prashant Shivaram, et al. "IMEX-Reg: Implicit-Explicit Regularization in the Function Space for Continual Learning." Transactions on Machine Learning Research.
>
> [5] Sarfraz, Fahad, Elahe Arani, and Bahram Zonooz. "Sparse coding in a dual memory system for lifelong learning." Proceedings of the AAAI Conference on Artificial Intelligence. Vol. 37. No. 8. 2023.
>
> [6] Gowda, Shruthi, Bahram Zonooz, and Elahe Arani. "Dual Cognitive Architecture: Incorporating Biases and Multi-Memory Systems for Lifelong Learning." Transactions on Machine Learning Research.
>
> I would strongly suggest the authors to extensively study the literature and critically re-evaluate the novelty and contributions of their study.

---

> > ### Author Response · Authors · 2025-09-08
> >
> > We thank the reviewer for the thoughtful feedback and for pointing us toward additional relevant literature. In response, we have substantially revised our positioning and related works section to better situate our contribution within the literature. Specifically, we would like to emphasize the following updates and clarifications:
> >
> > Positioning and Novelty: We have carefully re-evaluated the novelty of our approach given the suggested works (Section 1). Our key contribution is to bridge the gap between post-hoc model merging methods and existing in-training averaging techniques. While it is true that prior works have explored using just one auxiliary model or shown efficacy without a buffer, they alternatively tend to introduce additional complexity, either in the form of regularizers or costly per-iteration averaging (which in our experiments on practically sized large language models was unworkably inefficient). In contrast, our method offers a practical alternative: periodic, user-controlled averaging that requires no buffers, no auxiliary networks, and avoids the cost of per-iteration updates. Moreover, we show that in some cases, performing averaging less frequently actually yields substantially better performance, as demonstrated for CIFAR-100 and Food-101. This makes our approach low-overhead, simple to implement, and cost-effective, while still highly effective in mitigating forgetting. Furthermore, we extend evaluation beyond vision, showing that our approach is also effective when fully fine-tuning large language models (>1B parameters) on language tasks.
> >
> >
> > Related Literature: We have greatly expanded the related and cited works, including the works mentioned, as well as, additional related techniques (Sections 1 and 2), and expanded our discussion to critically compare their assumptions, costs, and mechanisms with our own.
> >
> >
> > Task Boundary Additional Experiments: We have added an implementation of SFA that does not rely on task boundaries and have added experiments (Appendix SFA Without Task Boundaries) that demonstrate the efficacy of our approach, addressing scenarios where task boundaries are unknown or unavailable.
> >
> >
> > CLS-ER and Baselines: As noted in our earlier response, we have now included CLS-ER as a baseline (Section 4, Figure 3) and provided a direct comparison in terms of both performance and storage overhead. We also contrast our method against CLS-ER and buffer-based approaches more generally, highlighting the differences in storage requirements and implementation complexity (Appendix Comparing Storage Overhead).
> >
> >
> > We hope these revisions and additions address the reviewer’s concerns. We believe our method makes a meaningful contribution by offering a uniquely simple and practical form of in-training model averaging, complementing the more complex approaches in the literature.
> > We are happy to further expand our discussion, include additional comparisons, or run further experiments. Please let us know if there are specific aspects you believe remain unaddressed, and we will make every effort to incorporate them.

---

### Review · Reviewer_oRyL · 2025-08-20

**Summary Of Contributions:**

The paper addresses catastrophic forgetting in continual learning, particularly in scenarios with large domain shifts between tasks. The authors propose Sequential Fine-tuning with Averaging (SFA), a method that periodically merges the parameters of the current model being fine-tuned with a checkpoint from previous tasks.

**Audience:**

Yes

**Broader Impact Concerns:**

/

**Claims And Evidence:**

Yes

**Requested Changes:**

please see weakness

**Strengths And Weaknesses:**

Strengths:
- Overall, the paper is easy to read.
- The proposed SFA method is simple, computationally efficient, and also effective.
- Experiments are conducted on both LLMs and image models.

Weakness:
- Although discussed by the authors, the proposed idea is conceptually close to prior work (e.g., WiSE-FT). The difference between the two may not constitute a substantial methodological advance and could be seen as incremental.
- For key results, tables would be clearer than figures for precise comparison.
- The hyperparameter 𝑝 appears to have a large impact on results, with very specific values (e.g., 0.21, 0.43, 1, 0.98, 0.64, 0.32, 0.96) in Figure 3. How can p be selected efficiently? If extensive tuning is required, the practicality and claimed efficiency of the method may be reduced.
- For extremely large models (e.g., 70B+), how feasible is frequent checkpoint storage and averaging?
Could SFA be combined with partial-parameter fine-tuning (e.g., LoRA) to reduce memory and compute requirements?

---

> ### Author Response · Authors · 2025-08-26
>
> We thank the reviewer for their comments and feedback. We address the listed Weaknesses in order:
>
> 1) While SFA builds on existing merging ideas, it uses p to actively control the forgetting-plasticity trade-off during fine-tuning, unlike many other merging methods. To this end, performance is non-incremental where domain shifts are large (e.g., language tasks), precisely where end-of-task merging (p=1, as in, e.g., WiSE-FT) often fails, while SFA (p<1) succeeds (Figure 2, SFA (p=0.25) retains 0.35 on GSM8K while WiSE-FT is only able to retain 0.01). As such, while also incorporating merging, SFA still consistently outperforms WiSE-FT.
> Furthermore, SFA is easy to implement for users already familiar with model merging, as it may only require a simple change to existing code bases for performance improvements..  As such, our work can be viewed as a generalization to WiSE-FT that particularly outperforms the former on language tasks.
>
> 2) We actually do provide tables for our main paper figures (See Tables 2-8 in the Appendix).
>
> 3) Thank you for pointing this out. We have added additional clarification to emphasize our recommendation for finding p based on our results.
>
> Currently, in our experiments, we treat p as a standard hyperparameter that we perform a sweep on with generally 3-5 values, and our graphs demonstrate this (e.g. Figure 1). This is what we currently promote as our recommendation for finding p given unique use cases and datasets. We also find that such a sweep provides good results without additional extensive tuning. Furthermore, it appears that among language tasks, p=0.25 seems to be a generally good value.
>
> We do believe that future work which focuses on more efficient selection of p would be very useful. Perhaps, given the range of unique tasks and data, the selection of p may be addressed through additionally training a model to infer optimal p range given tasks and current state.
>
> 4) In our current setup, we perform full fine-tuning and merge all parameters. While our method is generally cost efficient (requiring only the storage of 2 models for infrequent averaging given any number of tasks in a sequence), to save additionally we think that combining partial parameter fine-tuning and SFA would be optimal, especially given existing work that shows how combining partial parameter fine-tuning methods (e.g. LoRA) with final model averaging is effective [1].
>
> Thank you for this really interesting idea. While it is outside the scope of our current introduction to SFA, we believe partial parameter fine-tuning with SFA is a promising extension of our method that should still be effective. [1] have explored averaging models fully trained with LoRA on different tasks. As such, it is reasonable to assume that continuous averaging during training with LoRA, or generally partial parameter fine-tuning, would also work.
>
> [1] P. Plantinga, J. Yoo, A. Girma and C. Dhir, "Parameter Averaging Is All You Need To Prevent Forgetting," 2024 IEEE Spoken Language Technology Workshop (SLT), Macao, 2024, pp. 271-278, doi: 10.1109/SLT61566.2024.10832275.

---

### Decision · Action_Editor_NeU1 · 2025-11-16

**Recommendation:** Reject

**Additional Comments:**

Reviewers converged on the idea that the paper’s key claims are overstated given existing literature. The related work is incomplete and does not fairly cover the many closely related methods; as a result, the contribution is not well distinguished. While the authors attempted to address concerns during rebuttal and added some extra comparisons, reviewers appreciated the effort but found the additions insufficient and recommended rejection.

If you choose to resubmit, consider the points raised by reviewers, and please recalibrate the claims to reflect prior works, substantially expand and balance the related work (including recent in‑training/model‑averaging and rehearsal‑free CL), and expand evaluations against the most relevant baselines with transparent buffer/tuning details, ablations, and storage/compute analyses; also improve clarity of notation and tables/figures. Make claims match results: if the contribution is a careful empirical study rather than a new method, say so; otherwise show clear, controlled gains beyond the closest prior approaches. Addressing the points above would improve the new revision.

**Audience:**

Yes

**Audience Explanation:**

The question of how averaging frequency affects stability/forgetting is relevant, in general, especifically for the CL community. **The problem is not the topic, but that the current framing overstates the contribution relative to prior work, and the evidence does not yet support the stated claims.**

**Claims And Evidence:**

No

**Claims Explanation:**

All three reviewers questioned the claims/evidence mismatch in the paper’s positioning. The paper overstates its contribution relative to prior work and does not adequately situate itself in the literature. The current evidence does not convincingly demonstrate advances beyond approaches already used in the field for a couple of years now. Concretely, the introduction frames the method as addressing a **gap** (“To address this gap, we propose…”) and emphasizes the tunable in-training averaging frequency as a key differentiator (“Unlike approaches that merge continuously, SFA introduces a tunable averaging frequency parameter (p)…”), yet closely related in-training/online averaging with varying update frequencies has been studied in prior continual-learning work (including work cited in the paper). As written, the manuscript does not clearly distinguish what is new beyond these lines, nor does it provide sufficiently controlled comparisons details to justify the strength of the “SOTA/comparable to rehearsal” claims.

**Resubmission Of Major Revision:**

The authors may consider submitting a major revision at a later time.